# Bayesian estimation of Earth's climate sensitivity and transient climate response from observational warming and heat content datasets

Philip Goodwin[1], B. B. Cael[2]

[1]School of Ocean and Earth Science, University of Southampton, SO13 3ZH, UK
[2]National Oceanography Centre, Southampton, SO13 3ZH, UK

*Correspondence to*: Philip Goodwin (p.a.goodwin@soton.ac.uk)

**Abstract.** Future climate change projections, impacts and mitigation targets are directly affected by how sensitive Earth's global mean surface temperature is to anthropogenic forcing, expressed via the climate sensitivity (S) and transient climate response (TCR). However, the S and TCR are poorly constrained, in part because historic observations and future climate projections consider the climate system under different response timescales with potentially different climate feedback strengths. Here, we evaluate S and TCR by using historic observations of surface warming, since the mid-19[th] century, and ocean heat uptake, since the mid 20[th] century, to constrain a model with independent climate feedback components acting over multiple response timescales. Adopting a Bayesian approach, our prior uses a constrained distribution for the instantaneous Planck feedback combined with wide-ranging uniform distributions of the strengths of the fast feedbacks (acting over several days) and multi-decadal feedbacks. We extract posterior distributions by applying likelihood functions derived from different combinations of observational datasets. The resulting TCR distributions when using two preferred combinations of historic datasets both find a TCR of 1.5 (1.3 to 1.8 at 5-95% range) °C. We find the posterior probability distribution for S for our preferred dataset combination evolves from S of 2.0 (1.6 to 2.5) °C on a 20-year response timescale to S of 2. 3 (1. 4 to 6.4) °C on a 140-year response timescale, due to the impact of multi-decadal feedbacks. Our results demonstrate how multi-decadal feedbacks allow significantly higher upper bound on S than historic observations are otherwise consistent with.

## 1 Introduction

A key goal in climate science is to evaluate how sensitive global mean temperature anomaly is to radiative forcing from greenhouse gasses and aerosols (e.g. Knutti et al., 2017; IPCC, 2013). This sensitivity of climate may be explored by considering how a global surface temperature anomaly affects Earth's radiation balance. The effective climate feedback, $\lambda_{eff}$ (Wm$^{-2}$K$^{-1}$), expresses the how surface warming increases outgoing radiation at the top of the atmosphere. $\lambda_{eff}$ at some time $t$ is calculated from the total radiative forcing, $R_{total}$ (Wm$^{-2}$), the net top-of-atmosphere energy imbalance, $N$ (Wm$^{-2}$), and the global surface temperature anomaly, $\Delta T$ (K), via

$$\lambda_{eff}(t) = \big(R_{total}(t) - N(t)\big)/\Delta T(t)$$

(1)

where both $R_{total}$ and $\Delta T$ are defined as zero at some preindustrial state. The climate sensitivity at some time $t$, $S(t)$ in K, is
then defined as the radiative forcing for a doubling of $CO_2$, $R_{2 \times CO2}$, divided by $\lambda_{eff}(t)$,

$$S(t) = \frac{R_{2 \times CO2}}{\lambda_{eff}(t)} = \frac{R_{2 \times CO2} \Delta T(t)}{R_{total}(t) - N(t)}$$

(2)

S and $\lambda_{eff}$ may be evaluated from estimates of historic radiative forcing and observational constraints on $\Delta T$ and $N$, eqns. (1,
2); noting that Earth's energy imbalance, $N$, can be observationally constrained as a time-average through reconstructing the heat content changes in the Earth system dominated by the ocean (e.g. Cheng et al., 2017; Levitus et al., 2012).

Many previous studies evaluating S from historical observational data and radiative forcing estimates, eq. (2), have either calculated a single constant climate sensitivity (see Annan, 2015; Anan and Hargreaves, 2020; Bodman and Jones, 2016; Lewis
and Curry, 2014; Sherwood et al., 2020; Skeie et al, 2018; Otto et al., 2013; Nijsse et al., 2020), or have evaluated S for specific historic periods (e.g. Tokarska et al., 2020), acknowledging that the value for the specific historical period may not apply for all timescales into the future.

The assumption of a single constant S over time leads to uncertainties arising from model inadequacy (Annan, 2015), since
climate sensitivity may not be constant with time or across different response timescales (e.g. Rugenstein et al., 2020; Rohling et al., 2012; 2018; Goodwin, 2018; Knutti et al., 2017; Senior and Mitchell, 2000; Proistosescu and Huybers, 2017). There is also the possibility that, at any given time or timescale, the climate feedback may be different for different sources of radiative forcing, such as well mixed greenhouse gasses and volcanic aerosols (e.g. Marvel et al., 2015).

The aim here is to perform Bayesian probabilistic evaluations of both S and transient climate response (TCR in K), using observational constraints on global surface temperature and ocean heat content anomalies to constrain a model framework that includes time-varying climate feedbacks, eqns. (1, 2). Our estimates of S and TCR are independent of simulated warming responses in complex climate models (in contrast to estimates utilising complex model output via emergent constraints, e.g. Nijsse et al., 2020).


We utilise a numerical model that allows multiple climate feedbacks to each respond to radiative forcing over different timescales (Goodwin, 2018), such that $\lambda_{eff}$ to varies over time (eqns. 1,2). This study considers the instantaneous Planck feedback and two further timescales of climate feedback: a multiday feedback representing a selection of fast climate processes,

such as water vapour and clouds, and a multi-decadal climate feedback representing slower processes, such as the surface warming pattern effect. Generating a prior model ensemble with varying fast and multi-decadal climate feedback strengths, we extract three posterior ensembles using a Bayesian comparison to observational reconstructions. Each posterior ensemble applies a different combination of historic reconstructions of global surface temperature anomaly (either HadCRUT5 or HadCRUT5 without statistical infilling of geographically absent data [hereafter HadCRUT5 (no infill)]; Morice et al., 2021; Fig. 1a) and reconstruction of ocean heat content anomaly (either Cheng et al: Cheng et al., 2017; or NODC: Levitus et al., 2012; Fig. 1b). All our posterior ensembles are extracted using the additional constraints from HadSST4 (Kennedy et al., 2019) and Global Carbon Budget (le Quéré et al., 2018) for sea surface temperature and ocean carbon uptake anomalies respectively (see Supplementary Information).

## 2 Model of surface warming from time-varying climate feedback

Equation (1) considers surface warming via a single effective climate feedback response to total radiative forcing, where the effective climate feedback represents an aggregated response to multiple climate feedbacks to multiple sources of radiative forcing. Here, surface warming is modelled as an extended energy balance response to $i$ sources of radiative forcing by $j$ climate feedbacks operating over different response timescales (Goodwin, 2018),

$$\Delta T(t) = \left(1 - \frac{N(t)}{R_{total}(t)}\right) \sum_i \left[\frac{R_i(t)}{\lambda_{Planck} + \sum_j \lambda_{i,j}(t)}\right]$$

$$(3)$$

The $j$ combinations of climate feedbacks processes considered here are:

(1) $\lambda_{fast}$, the combined fast feedbacks operating over response timescales approximately linked to the residence timescale of water vapour in the atmosphere (van der Ent and Tuinenberg, 2017), including clouds, water vapour-lapse rate, snow and sea-ice surface albedo; and

(2) $\lambda_{multidecadal}$, the combined feedbacks operating over a multi-decadal timescale that may, for example, be linked to a surface warming pattern adjustment (e.g. Andrews et al., 2015).

Note that slow climate feedbacks with timescales longer than multi-decadal are not explored here, since the historical records of temperature and heat content changes do not extend long enough to offer a reliable constraint on processes acting on such long timescales. Also, the snow and ice albedo feedback has a timescale longer than the atmospheric water vapour residence timescale, but is included in $\lambda_{fast}$ here as the timescale snow and sea-ice responds significantly faster than multi-decadal timescales. The sign convention adopted has positive overall $\lambda_{eff}$, such that negative $\lambda_{fast}$ and $\lambda_{multidecadal}$ are amplifying.

The WASP model starts simulations at year 1700 by default (e.g. Goodwin, 2018), with different sources of radiative forcing
defined from some time after that date. While the observational constraints used in this study start in year 1850, the model
state in 1850 is affected by radiative forcing received prior to that date. Therefore, this study imposes radiative forcing on the
WASP model prior to 1850. The $i$ sources of radiative forcing used in eq. (3) are:

(1) Atmospheric $CO_2$ forcing, calculated from $CO_2$ concentrations using $R_{CO2} = a_{CO2}\Delta ln CO_2$ after IPCC (2013);

(2) Combined forcing from other well mixed greenhouse gases, $R_{WMGHG}$, including methane, nitrous oxides each calculated
from concentrations after Etminan et al. (2016) (see Supplementary Information), and halocarbons after IPCC (2013);

(3) Combined direct and indirect anthropogenic aerosol forcing, linked annual aerosol emission rates (Myhre et al., 2013;
Smith et al., 2018, see Supplementary Information);

(4) Volcanic aerosol radiative forcing, calculated after 1850 from volcanic Aerosol Optical Depth (AOD) using $R_{volcanic} = -(19 \pm 0.5)AOD$ (Gregory et al., 2016) and before 1850 from the global radiative forcing timeseries used in the Reduced
Complexity Model Intercomparison Project (RCMIP) phase 1 (Nicholls et al., 2020), with identical relative uncertainty
imposed both pre and post 1850;

(5) Solar forcing; and

(6) Internal variability in Earth's energy imbalance, imposed using AR1 noise with coefficients chosen to approximate the
properties of monthly and yearly average noise from Trenberth et al. (2014).

The equations WASP uses to evolve climate feedback over time are presented in Goodwin (2018), and discussed here in the
Supplementary Information. Briefly, when radiative forcing from source $i$ is not increasing in magnitude between times $t - \delta t$
and $t$, $|R_i(t)| \leq |R_i(t - \delta t)|$, the $j$th combination of climate feedback processes evolves according to (see Supplementary
Information),

$$\lambda_{i,j}(t) = \lambda_{i,j}(t - \delta t) + \left(\lambda_j^{equil} - \lambda_{i,j}(t - \delta t)\right)\left(1 - \exp\left(\frac{-\delta t}{\tau_j}\right)\right)$$

(4)

However, when radiative forcing from source $i$ is increasing in magnitude, $|R_i(t + \delta t)| > |R_i(t)|$, climate feedback $\lambda_{i,j}$
evolves from $t$ to $t + \delta t$ according to (see Supplementary Information),

$$\lambda_{i,j}(t) = \left|\frac{R_i(t - \delta t)}{R_i(t)}\right|\left(\lambda_{i,j}(t - \delta t) + \left(\lambda_j^{equil} - \lambda_{i,j}(t - \delta t)\right)\left(1 - \exp\left(\frac{-\delta t}{\tau_j}\right)\right)\right)$$

(5)

Thus, from eqns. (3), (4) and (5), any additional radiative forcing acts instantaneously at the Planck feedback in the first time-
step it is applied, and then evolves over the e-folding response timescales $\tau_j$ towards the equilibrium climate feedback,

$\lambda_{equilibrium} = \lambda_{Planck} + \lambda_{fast}^{equil} + \lambda_{multidecadal}^{equil}$. Supplementary Figure S7 shows how climate feedback evolves over time in response to an idealised radiative forcing using equations (4) and (5). Since eq. (5) is applied separately for each of the $i$ sources of radiative forcing, the framework used here allows different values of climate feedback at any point in time for each source of radiative forcing.

This model of climate feedbacks responding to imposed radiative forcing over multiple response timescales, eqns. (3), (4) and (5), produces a time-evolving effective climate feedback, (1), and time-evolving climate sensitivity, (2), in response to a prescribed forcing scenario. Here, the transient climate response, TCR, is calculated as the 20-year average warming centred at the year of $CO_2$ doubling for a scenario with a 1 per cent per year rise in $CO_2$ and no other forcing (hereafter: 1pctCO2 scenario).


### 3 Generation of the prior and posterior ensembles

We generate probabilistic prior and posterior model ensembles with varied model input parameters using Bayes' theorem. The joint posterior probability that the climate system parameters $X$ have a specific set of values $X'$ given background information $I$ and observations of the climate system $\{obs\}$, $prob(X = X'|\{obs\}, I)$, is expressed using Bayes' theorem,


$$prob(X = X'|\{obs\}, I) \propto prob(\{obs\}|X = X', I) \times prob(X = X'|I)$$

(6)

where:

(1) $prob(X = X'|I)$ is the joint prior probability that $X = X'$ for climate system parameter values (Supplementary Table S1; Fig. 2 solid lines for $\lambda_{Planck}$, $\lambda_{fast}^{equil}$ and $\lambda_{multidecadal}^{equil}$); and

(2) $prob(\{obs\}|X = X', I)$ is known as the likelihood function and expresses the probability of obtaining the observations in $\{obs\}$ for the given joint parameter values $X = X'$ and background information $I$. Here, this is estimated from where the simulated model observables for $X = X'$ and $I$ lie on the probability distributions for the real observables (Supplementary

Table S2).

Here, we use large ensemble simulations of the Warming Acidification and Sea level Projector (WASP) model (Goodwin, 2016), adopting the updated version of Goodwin (2018) with explicitly time-evolving climate feedbacks (eqns. 3, 4 and 5; see Supplementary Information). This version of WASP does not contain a single parameter for S or $\lambda_{eff}$ at some time $t$, eqns.

(1, 2). Instead, the values of S and $\lambda_{eff}$ emerge over time in the model in response to the forcing scenario from a combination of multiple prescribed climate system parameters (eqns. 3, 4, 5). The WASP model contains a 5-box representation of ocean

heat and carbon uptake, with an ocean circulation that is varied between ensemble members but remains constant in time within each ensemble member (Supplementary Table S1).

We form a prior model ensemble where a total of 25 model input parameters independently varied between simulations (Supplementary Table S1), to represent the prior climate system parameter distribution $X$, eq. (6). Five of the input parameters within $X$ describe how climate feedback responds to an imposed radiative forcing ($\lambda_{Planck}$, $\lambda_{fast}^{equil}$, $\lambda_{multidecadal}^{equil}$, $\tau_{Fast}$ and $\tau_{Slow}$) with a 6th input parameter (the radiative forcing coefficient for $CO_2$) converting this climate feedback to climate sensitivity (Supplementary Table S1, eq. 2). The $\lambda_{Planck}$ parameter is randomly varied from normal distribution (Fig. 2a,

black solid line), while the $\lambda_{fast}^{equil}$ and $\lambda_{multidecadal}^{equil}$ parameters are randomly varied from uniform distributions (Fig. 2b,c, black solid lines) reflecting the degree of assumed prior knowledge of their values (Supplementary Information).

A further thirteen of the 25 model input parameters varied within $X$ relate to uncertainty in historic radiative forcing

(Supplementary Table S1). The WASP model is historically forced until 2014 (following the ssp585 scenario thereafter: O'Neill et al., 2016) with atmospheric concentrations of greenhouse gasses; direct and indirect radiative forcing from anthropogenic aerosols; radiative forcing from volcanic aerosols; and solar forcing (see Supplementary Information). The radiative forcing from each component (aside from solar forcing) is varied between simulations in the prior ensemble (Supplementary Table S1) to approximate historic uncertainty (Myhre et al., 2013; Etminan et al., 2016; Smith et al., 2018;

Gregory et al., 2016).

Normal input distributions (Supplementary Table S1) are used to represent historic uncertainty in: the radiative forcing sensitivity to greenhouse gas concentrations (Myhre et al., 2013; Etminan et al., 2016); the direct radiative forcing sensitivity to anthropogenic aerosol emissions for six separate aerosol types (Myhre et al., 2013), and the radiative forcing sensitivity to

volcanic aerosol optical depth (Gregory et al., 2016). However, a skew-normal input distribution is used to represent historic uncertainty in the indirect radiative forcing from anthropogenic aerosols (Supplementary Table S1), since there is a long tail of possibly strong-negative radiative forcing from this effect (IPCC, 2013). The input distributions of direct and indirect aerosol radiative forcing coefficients together produce a broad and skewed prior distribution of total recent radiative forcing from aerosols (Fig. 3, black solid and dotted lines: shown for year 2014) with similar mean to the best estimate of recent aerosol

radiative forcing from IPCC (2013) Assessment Report 5 (Fig. 3, compare black to light blue with IPCC AR5 estimate shown for year 2011).

We generate three prior ensembles containing from $2.1 \times 10^9$ to $4.6 \times 10^9$ ensemble members. In each prior ensemble the 25 input parameters independently varied such that the relative frequency distributions of each input parameter are set to the

assumed prior probability distribution, $prob(X = X'|I)$ in eq. (6) (Supplementary Table S1; Fig. 2 solid lines for $\lambda_{Planck}$, $\lambda_{fast}^{equil}$ and $\lambda_{multidecadal}^{equil}$). Observational tests from three combinations of historic datasets are then used to form a likelihood function and extract a subset of the prior ensemble simulations into the posterior ensembles (Supplementary Table S2).

There are $n = 12$ observational constraints within $\{obs\}$ (Supplementary Table S2). The probability of obtaining the $k^{th}$
observational constraint given $X = X'$ and $I$ is calculated assuming Gaussian uncertainty in the observable (e.g. Annan and Hargreaves, 2020),

$$prob(\{obs\}_k|X = X',I) \propto e^{\frac{-(\mu_k-x_k)^2}{2\sigma_k^2}}$$

(7)

where $\mu_k$ and $\sigma_k$ are the observational mean and standard deviation uncertainty of observable $k$ (Supplementary Table 2), and
$x_k$ is the simulated value of the observable for $X = X'$ and $I$. To calculate the overall probability of obtaining all $n$ observational constraints within $\{obs\}$ given $X = X'$ and $I$, we multiply the probabilities for all $\{obs\}_k$,

$$prob(\{obs\}|X = X',I) = \prod_{k=1}^{n} prob(\{obs\}_k|X = X',I)$$

(8)

Three different ensembles are generated using different combinations of surface temperature (HadCRUT5 and HadCRUT5
(no infilling): Fig. 1a) and heat content (Cheng et al. and NODC: Fig. 1b) datasets to construct the likelihood function that acts as a constraint on the posterior (eq. 6). These model ensembles are termed  HadCRUT5 & Cheng et al.; HadCRUT5 & NODC.; and HadCRUT5 (no infilling) & Cheng et al. (Supplementary Table S2). The preferred combination of observational datasets is HadCRUT5 & Cheng et al., as these represent the most up to date methodologies for their respective temperature (Morice et al., 2021) and heat content (Cheng et al., 2017) reconstructions. The other dataset combinations are included to assess the
sensitivity of our method to different heat content datasets (HadCRUT5 & NODC) and the sensitivity of our findings to the statistical infilling of missing data (HadCRUT5 (no infill) & Cheng et al.). It is noted that most other temperature datasets now reconstruct similar historic global mean temperature anomalies to HadCRUT5 (e.g. see Morice et al. 2021).

For each of the three posterior ensembles, corresponding to different dataset combinations, the probability of a prior simulation being included within the posterior ensemble is proportional to $prob(\{obs\}|X = X',I)$, eqn. (8): a simulation is accepted into the posterior ensemble if the value of $prob(\{obs\}|X = X',I)$, assessed using (8), is greater than a number randomly drawn between 0 and some number greater than or equal to the maximum value of $prob(\{obs\}|X = X',I)$ achieved in that prior ensemble.


We adopt a normal prior distribution for $\lambda_{Planck}$, informed by Earth's global mean surface temperature (Jones and Harpham, 2013) and radiation budget (Trenberth et al., 2014) (Fig. 2a, solid black line). We adopt uniform prior distributions of $\lambda_{fast}^{equil}$ and $\lambda_{multidecadal}^{equil}$ (Fig. 2b,c solid black lines), thus assuming that any value within the boundaries is equally likely before we consider the observations, $\{obs\}$ (eq. 6). Our boundaries for the uniform distributions of $\lambda_{fast}^{equil}$ and $\lambda_{multidecadal}^{equil}$ are set wide enough such that the posterior distributions are not significantly affected by the boundaries (Fig. 2, red and blue), but narrow enough such that the problem is computationally tractable. The distribution for $\lambda_{multidecadal}^{equil}$ is centred at 0, such that no prior assumption is made as to whether multi-decadal feedbacks will amplify or dampen future warming (Fig. 2).

## 4 Results

The three prior and posterior ensembles generated range in size: a total of 1764 simulations are accepted into the HadCRUT5 & Cheng et al. posterior ensemble from an initial prior ensemble of $4.6 \times 10^9$ simulations; a total of 2997 simulations are accepted into the HADCRUT5 & NODC posterior ensemble from an initial prior ensemble of $2.7 \times 10^9$ simulations; and 9190 simulations are accepted into the HadCRUT5 (no infill) & Cheng et al. posterior ensemble from an initial prior ensemble of $2.1 \times 10^9$ simulations. A smaller fraction of the prior simulations are accepted into the posterior ensembles that use likelihood function terms, $prob(\{obs\}_k | X = X', I)$ in eq. (7), with smaller observational uncertainty, $\sigma_k$ (Supplementary Table S2).

The posterior distributions of climate feedback terms are similar for both ensembles constrained by the HadCRUT5 dataset (HadCRUT5 & Cheng et al., and HadCRUT5 & NODC), revealing that the Planck feedback, fast feedback and multi-decadal feedback strengths are insensitive to the choice of ocean heat content dataset used within the likelihood function (Figure 2a,b,c compare red and grey). The Planck feedback has posterior distributions in the range $\lambda_{Planck} = 3.3 \pm 0.1$ Wm$^{-2}$K$^{-1}$ for both ensembles (mean ± standard deviation: Fig. 2a, red and grey).

A strong compensatory link between fast and multi-decadal feedback strengths emerges in the posterior ensembles, with the HadCRUT5 & Cheng et al. ensemble revealing a best fit relationship of $\lambda_{fast}^{equil} = -1.59\lambda_{multidecadal}^{equil} - 2.51$, with R$^2$=0.92 (Fig. 2d). The posterior distributions for fast and multi-decadal climate feedback strengths are bimodal in the HadCRUT5 & Cheng et al. and HadCRUT5 & NODC ensembles (Fig. b,c, red and grey), corresponding to one observation consistent region with weaker amplifying fast feedback ($\lambda_{fast}^{equil} \sim -0.6$ Wm$^{-2}$) and strong amplifying multidecadal feedback ($\lambda_{multidecadal}^{equil} \sim -1.7$Wm$^{-2}$) , and another observation consistent region with very strong amplifying fast feedback ($\lambda_{fast}^{equil} \sim -2.2$Wm$^{-2}$) and damping multidecadal feedback ($\lambda_{multidecadal}^{equil} \sim +1$Wm$^{-2}$) (Fig. 2d, shown for the HadCRUT5 & Cheng et al. ensemble), noting that the sign convention used implies amplifying feedback from negative $\lambda$. This bimodality, with an unfavoured region

around $\lambda_{multidecadal}^{equil} \sim 0$ (Fig. 2), is consistent with effective climate feedback changing over the historic period (e.g. Gregory et al., 2019), since $\lambda_{multidecadal}^{equil} = 0$ would correspond with a constant value of $\lambda_{eff}$ over the entire historic period. The bimodality in the $\lambda_{fast}^{equil}$ and $\lambda_{multidecadal}^{equil}$ posterior distributions is not seen in the ensemble constrained by the temperature

reconstruction without statistical infilling (HadCRUT5 (no infill) & Cheng et al.), which instead has broader single-peak distributions (Fig. 2b,c blue).

## 4.1 The Climate Sensitivity and Transient Climate Response

S is analysed by forcing the four posterior ensembles with an instantaneous step-function quadrupling of atmospheric $CO_2$

(hereafter: 4x$CO_2$ scenario) and applying eq. (2) with 11-year averages. The value of S changes over time (Figs. 4,5) as the fast and multi-decadal climate feedbacks evolve in response to the imposed radiative forcing (eqns. 3, 4, 5).

For each combination of datasets used, S is best constrained from the historic observational reconstructions on 20-year timescale (Figs. 4,5a, Table 1). These 20-year response timescale S estimates are also similar between different dataset

combinations: varying from 2.1 °C (1.6 to 2.5 °C at 90% range from 5th to 95th percentiles) for the HadCRUT5 and Cheng et al. dataset combination to 2.1 °C (1.7 to 2.6 °C) for the HadCRUT5 & NODC dataset combination.

The distributions see a general increase in S out to 50-year, 100-year and 140-year timescales, with greater uncertainty (Figs. 4,5; Table 1) due to the uncertainty in how multi-decadal climate feedback will evolve (Fig. 2). The TCR is analysed by

forcing our posterior ensembles with a 1pct$CO_2$ scenario and recording the surface warming for each ensemble member for the 20-year average centred on the year in which $CO_2$ reaches twice its initial value (Fig. 6; Table 1). Our analysis reveals a TCR of 1.5 (1.3 to 1.8 at 90% range) °C when constrained by the HadCRUT5 temperature reconstruction with either ocean heat content dataset (Table 1).

## 4.2 Variation in the posterior model ensembles
The observational records provide constraints on the parameters of the posterior ensembles that manifest not only as posterior distributions for these parameters but also as relationships between them, as well as between model parameters and key model outputs of interest (such as S(t)). While the correlation structure of the 25 parameters' joint posterior distribution is generally quite complex, some key structures emerge that indicate how S and TCR uncertainties might be reduced. This method of

analysing variation, and simplifying the degrees of freedom of variation, in large data-constrained efficient model ensembles may ultimately help explore parameter space in more complex Earth system models.

### 4.2.1 Correlations of model parameters and outputs

We assemble the three observationally-consistent ensembles into a single meta-ensemble, where each model realization is weighted inversely to the number of members in its individual ensemble such that each of the three observational combinations is weighed equally (henceforth all analyses in this section are weighted, i.e. weighted correlations, weighted principal component analysis, and weighted stepwise regression). We then first examine the correlations between individual model parameters. We find three strongly correlated groups of model parameters (Supplementary Figure S1). First, the $\lambda_{multidecadal}^{equil}$ and $\lambda_{fast}^{equil}$ feedback parameters are strongly compensating ($\rho = -0.95$) and the $\lambda_{Planck}$ feedback is also fairly well-correlated with these ($\rho = 0.49$ and $-0.56$ respectively). Second, ratio 1 (the ratio of global near-surface warming to global sea surface warming at equilibrium) and ratio 2 (the ratio of global whole-ocean warming to global sea surface warming at equilibrium) parameters strongly compensate ($\rho = -0.85$), indicating the ratio of near-surface warming to global whole-ocean warming is tightly constrained by these datasets. Finally, all of the greenhouse gas and aerosol sensitivities are well-correlated, with $|\rho| \geq 0.4$ (except for the aerosol indirect effect). None of these are surprising as they reflect the primary constraints of the observations, i.e. ocean, near-surface warming, and radiative forcing histories, but the former does indicate that a better-constrained fast feedback parameter would directly reduce uncertainty on multidecadal feedbacks and thereby S on multidecadal and centennial timescales. Model outputs are in general correlated in expected fashions with each other and with model parameters. TCR is well-correlated with S on all timescales (20, 50, 100, and 140 years), and S on timescales greater than 20 years are all well-correlated, whereas $S_{20}$ is only weakly correlated with these as it is controlled by other feedback parameters. We therefore focus on TCR, $S_{20}$, and $S_{100}$ hereafter. S and TCR are, as expected, very strongly correlated with the feedback parameters and also appreciably correlated with greenhouse gas and aerosol sensitivity parameters, but weakly correlated with most other model parameters.

### 4.2.2 Principle components

Correlations between model parameters' posteriors imply that the dimensionality of the parameter space can be reduced and that the observational constraints collapse the posterior solution into a parameter space with fewer degrees of freedom. Principal Component Analysis, PCA (Jolliffe, 1986; n.b. we do not describe the method here as it is well-described in many textbooks such as Jolliffe,1986) is a straightforward, ubiquitous means to identify these degrees of freedom, and is justifiable here in the absence of strongly nonlinear model equations and given the Gaussian or near-Gaussian likelihoods and priors.

We perform a PCA on the model parameters' joint posterior; the results are presented in Figure 7. In the scree plot (fig. 7a) there is an obvious break point at the fifth principal component (PC), indicating the first five PCs are interpretable and the remaining are unstructured variations (Cattell, 1966). These PCs are shown in fig. 7b-f, with loadings of only the parameters with the absolute value of the loading >0.25 shown (full PCs are shown in Supplementary Figures S2-S6 for completeness). The first three of these PCs are dominated by the fast and multidecadal feedbacks and the sensitivity of radiative forcing to $CO_2$ and two aerosols ($SO_x$ and $NH_3$) (Fig. 7b-d). The fourth and fifth PC are dominated by oceanic factors (Fig. 7e,f): the

timescales of the multidecadal feedback and the ventilation of different ocean fractions, the buffered carbon inventory, and the warming ratios of near-surface to sea surface and sea surface to whole-ocean warming. Altogether these PCA results suggest that the observational constraints used herein collapse the 25 model parameters around a five-dimensional subspace, and that these five dimensions reflect the balance between the effects of climate feedbacks, greenhouse gases, and aerosols on atmospheric and oceanic warming, as well as the structure of the large-scale ocean circulation.

Note also there are numerous ways to quantify the number of interpretable or meaningful PCs resulting from a PCA (Jackson, 1993); the first five PCs we focus on here explain 60% of the total variance in the dataset, but the decisive break in the scree plot (fig. 7a) indicates strong evidence that these PCs are qualitatively different than the remaining PCs 6-25. We interpret the remaining variance in the data as reflective of the large amount of parametric uncertainty left in these models beyond what the observations herein can constrain, attesting to the importance of large ensemble simulations as employed here for quantifying uncertainty in S and TCR.

### 4.2.3 Stepwise regression

It is also of interest to what extent the model outputs are directly predictable from or explicable by the individual model parameters and/or PCs. Given the roughly Gaussian and linear model equations, multilinear regression is a suitable approach to identifying these links; in particular stepwise regression (Draper & Smith, 1981) follows an automatic procedure of including and removing explanatory variables from the model fit to identify an optimal combination. We perform stepwise regression to predict the model outputs from the model parameters and/or the first , (de)selecting explanatory variables using the Bayesian Information Criterion (Schwartz, 1978) and also including interactions between model parameters (i.e. their products).

We find $S_{100}$ to be significantly a function of all of PC1-5 and their interactions, with an $R^2 = 0.50$. While this is not an especially good fit, it is 83% of the variance in the model parameters explained by these PCs, i.e. almost all of the model parameter variance these PCs explain directly translates to explained variance in $S_{100}$. In combination with the PCA results, this suggests the observations used here collapse the model parameters around five degrees of freedom, and that $S_{100}$ is proportional to these degrees of freedom and their interactions, with the remaining variance in $S_{100}$ due to the remaining variance in the model parameters. This implies that the observational constraints used here directly constrain $S_{100}$ in our modeling approach, with very little information lost through constraining model parameters. In contrast, $S_{20}$ and TCR are more poorly predicted from these PCs ($R^2 = 0.37$ and $0.19$ respectively).

We also performed stepwise regression of model outputs against the 25 model parameters. We found $S_{100}$ to be a significant function of only nine model parameters (the three feedback parameters, the multidecadal feedback timescale, and the sensitivities to CO2, SOx, aerosol indirect forcing, VOC, and N2O), but very well-predicted by these parameters ($R^2 = 0.86$). $S_{20}$ was even better predicted ($R^2 = 0.96$) by a similar suite of parameters (exchanging sensivities to aerosol indirect forcing,

VOC, and N$_2$O with sensitivity to CH$_4$). This implies both that S is not strongly dependent on the other parameters in the model used here, and also that there is a large amount of variation in S that can be reduced by better constraining these parameters. In contrast, TCR is sensitive to more model parameters (fifteen) but is also similarly predictable from these ($R^2 = 0.95$).

## 4.3 Choice of priors and the sensitivity of results

The prior distributions for climate feedback terms adopted here (Figure 2a-c, black; Supplementary Table S1) will impact the posterior distributions for climate feedback terms, TCR, S$_{20}$, S$_{50}$. S$_{100}$ and S$_{140}$ (Table 1), but by how much? Our prior distribution for $\lambda_{Planck}^{equil}$ is chosen as normal with mean 3.3 Wm$^{-2}$K$^{-1}$ and standard deviation ±0.2 Wm$^{-2}$K$^{-1}$ since we have high confidence in the value from observational evidence (Jones and Harpham, 2013; Trenberth et al., 2014). However, the prior distributions for $\lambda_{fast}^{equil}$ and $\lambda_{multidecadal}^{equil}$ are uniform over broad ranges (Fig. 2b,c, black) to reflect our initial ignorance in their values (before applying observational constraints, Supplementary Table S2). Both prior distributions for $\lambda_{fast}^{equil}$ and $\lambda_{multidecadal}^{equil}$ have minimum values of -3.0 Wm$^{-2}$K$^{-1}$, chosen to be just less in magnitude than, and opposite in sign to, the mean in the prior for $\lambda_{Planck}^{equil}$, because we know that the total feedback must be positive on any timescale. The maximum value for $\lambda_{multidecadal}^{equil}$ is chosen to be +3.0 Wm$^{-2}$K$^{-1}$ so that the prior distribution for multidecadal feedbacks is symmetric around 0, and multidecadal feedbacks have an equal prior likelihood of amplifying or damping surface warming. The range of $\lambda_{fast}^{equil}$ is chosen to have maximum of +1.0 Wm$^{-2}$K$^{-1}$ to maximise computational efficiency, as this is higher than the maximum values found in the posterior ensembles (Figure 2c, dotted lines) so there is no need to extend the distribution further.

These are not the only prior distributions that could have been chosen, and the scatter relationship in Fig 2d shows that if we constrain one of either $\lambda_{multidecadal}^{equil}$ of $\lambda_{fast}^{equil}$ then we also constrain the other term. Sherwood et al. (2020) use evidence from a range of sources to justify a Gaussian distribution for the climate feedback due to the pattern effect that amplifies warming over several decades by 0.5 Wm$^{-2}$K$^{-1}$ with a 90% confidence interval of 0 to 1 Wm$^{-2}$K$^{-1}$. This is equivalent in this study to a normal distribution for $\lambda_{multidecadal}^{equil}$ with mean -0.5 Wm$^{-2}$K$^{-1}$ with a 90% confidence range of -1.0 to 0.0 Wm$^{-2}$K$^{-1}$ (Fig. 8, c purple), noting the change in sign convention relative to Sherwood et al. (2020). Since the evidence used by Sherwood et al. (2020) to justify this distribution does not contain the observational constraints used within the likelihood function in this study (Supplementary Table S2), we are free to adopt the Sherwood et al (2020) distribution as an alternative prior for $\lambda_{multidecadal}^{equil}$ (Figure 8c, purple). Here, the impact of this alternative prior on our results is explored by weighting the posterior simulations in the HadCRUT5 & Cheng et al. ensemble according to where their $\lambda_{multidecadal}^{equil}$ values fit within the Sherwood et al. (2020) prior distribution for multidecadal climate feedback relative to the uniform prior distribution (Fig. 8c, compare purple and black). Adopting this alternative prior on $\lambda_{multidecadal}^{equil}$ (Fig. 8c, purple) does indeed constrain the posterior distributions for both $\lambda_{multidecadal}^{equil}$ and $\lambda_{fast}^{equil}$ (Fig. 8, compare light blue to red). This has only a minor impact on TCR and climate sensitivity

on a 20-year timescale (Table 1), bur does greatly reduce uncertainty in posterior climate sensitivity on 100 and 140-year timescales (Table 1): changing $S_{140}$ from 2.3 (1.6 to 4.2 at 66% confidence) K with uniform prior to 2.4 (2.0 to 2.8) K with Sherwood et al. (2020) prior on multidecadal feedback. This reduction in confidence interval should be expected since the prior for $\lambda_{multidecadal}$ has been significantly reduced in range (Fig. 8c, compare black and purple). Full analysis of the impact of alternative priors is reserved for further study.

## 5. Discussion

Many studies have combined reconstructions of surface temperature and ocean heat uptake with estimates of radiative forcing to calculate the effective climate feedback and/or transient climate response during the historic period (e.g. Annan, 2015; Anan and Hargreaves, 2020; Bodman and Jones, 2016; Lewis and Curry, 2014; Skeie et al, 2018; Otto et al., 2013; Tokarska et al., 2020). However, climate feedback strengths evolve over time in complex climate models (e.g. Andrews et al., 2015), indicating that climate sensitivity values obtained from historic observations may not apply into the future.

This study applies the historic observational record (Supplementary Table S2) and estimates of historic radiative forcing (Figure 3; Supplementary Table S2) to constrain how climate sensitivity evolves on different response timescales (Figs. 4,5), utilising a model of independent climate feedback terms that respond to forcing over instantaneous (Planck), fast (several days) and multi-decadal timescales (eqns. 2,3,4). A Bayesian approach is adopted, where uniform prior probability distributions are applied for the fast and multi-decadal climate feedbacks (Fig. 2, Supplementary Table S1). Different temperature and ocean heat content observational datasets (Supplementary Table S2, eqns. 6,7,8) are applied to extract posterior probability distributions for climate feedbacks (Fig. 2) and other model properties (e.g. related to aerosol radiative forcing Fig. 3). We then use these posterior probability distributions to evaluate climate sensitivity (S) and transient climate response (TCR) from $4xCO_2$ and $1pctCO_2$ forcing scenarios respectively.

Our estimates of S on a 20-year timescale is directly comparable to estimates of climate sensitivity made from historical constraints (e.g. Otto et al., 2013; Lewis and Curry, 2014), without explicitly considering the impact of additional slower (including multi-decadal) climate feedbacks that may not have had time to equilibrate in the present day.

Our estimates of S on 100-year and 140-year timescales are directly comparable to the climate sensitivity estimates evaluated in complex climate model simulations from simulations lasting order 100 years, for example using the Gregory et al. (2004) method. Note that additional slow feedbacks not considered here, acting from many decades to millennia, may affect how our estimates are comparable to estimates of climate sensitivity from the palaeo-record where any longer feedbacks have been treated as radiative forcing (e.g. Rohling et al., 2012; 2018).

We find that the HadCRUT5 (Morice et al., 2021) temperature reconstruction implies a larger S and TCR than HadCRUT5 (no infill) (Figs. 4,5,6; Table 1), demonstrating the importance of statistical infilling of geographical areas absent in historic when constraining future warming (Cowtan and Way, 2014). The Cheng et al. (2017) ocean heat content reconstruction implies similar S to the NODC reconstruction (Fig. 3,5; Table 1), showing the insensitivity of our results to these different heat content reconstructions (Fig. 1b). The different heat content datasets make almost no impact on TCR (Fig. 6; Table 1), which may be expected when considering that a larger historic heat content also implies larger heat uptake on a 1pctCO$_2$ scenario and this balances any warming impact of a larger S. An alternative narrower prior distribution for multidecadal climate feedback (Sherwood et al., 2020) reduces uncertainty in our posterior estimates of S on longer, 100 and 140-year, timescales, but has little impact on TCR, or S on a 20-year timescale (Table 1, Fig. 8).

Our method constrains S over multiple response timescales (Fig. 4; Table 1). Our constraints on S over a 100-year and 140-year response timescales (S$_{100}$, S$_{140}$: Table 1) are directly comparable to previous reviews of climate sensitivity in the literature in AR5 (IPCC, 2013) and Sherwood et al. (2020). The IPCC (2013) AR5 estimate of ('effective' or 'equilibrium') climate sensitivity has a 66% (or better) likelihood range of 1.5 to 4.5 K (IPCC, 2013), while the recent Sherwood et al. (2020) Bayesian review has a narrower baseline 17$^{th}$ -83$^{rd}$ percentile (66%) range of 2.6 to 3.6 K. The Sherwood et al. (2020) range removes both the lower portion of the IPCC climate sensitivity likely (66% likelihood or better) range (from 1.5 to 2.5 K) and the upper portion (from 3.7 to 4.5 K), suggesting a similar best estimate but with reduced uncertainty than IPCC (2013).

Our posterior 66% range for S$_{140}$ (of 1.6 to 4.2 K for our preferred HadCRUT5 & Cheng et al. ensemble) is in very good agreement with the equivalent IPCC (2013) range (Table 1), and broader than the recent Sherwood et al. (2020) range. Both Sherwood et al. (2020) and this study apply Bayesian approaches to constrain effective climate feedback, $\lambda_{eff}$, and use this constraint on $\lambda_{eff}$ to then constrain S. Our broader range compared to Sherwood et al. (2020) may arise from differences in our Bayesian approaches. Firstly, Sherwood et al. (2020) considers additional sources of evidence, for example from palaeoclimate reconstructions, that may narrow their range of climate sensitivity relative to ours. Secondly, our methodology includes a model of climate feedback that is explicitly allowed to evolve over different response timescales (Fig. 2; eqns. 1-5; Supplementary Information), with equal prior weighting given to amplifying and damping feedback evolution over multidecadal timescales (Fig. 2b). This time evolution in $\lambda_{eff}$ thus allows S to also evolve over different response timescales (Fig. 3, Table 1), and prevents our approach from over-constraining S on multidecadal and century timescales from historical datasets that record only the decadal responses to recent anthropogenic forcing. When a narrower prior distribution for multidecadal climate feedback from Sherwood et al. (2020) is adopted within our approach, our posterior S$_{140}$ decreases in range (2.0 to 2.8 K at 66% confidence: Table 1), but to lower values than Sherwood et al. (2020). It should be noted that additional slow feedbacks acting on longer timescales (century and longer) may allow climate sensitivity to evolve further (e.g. Rohling et al., 2021; 2018), but are not considered in our methodology.

**Code availability:** The WASP model code used here is available for download at http://doi.org/10.5281/zenodo.4639491

450     **Data availability:** The datasets used (Supplementary Table S2) are publicly available: HadCRUT5 is available at
        https://www.metoffice.gov.uk/hadobs/hadcrut5/data/current/download.html. NODC is available at
        https://www.nodc.noaa.gov/OC5/3M_HEAT_CONTENT/. Cheng et al. is available at
        http://159.226.119.60/cheng/images_files/IAP_OHC_estimate_update.txt. HadSST4 is available at
        https://www.metoffice.gov.uk/hadobs/hadsst4/data/download.html. The Global Carbon Budget 2018 data is available at
455     https://doi.org/10.18160/gcp-2018.

**Code availability:** The WASP model code used here is available for download at http://doi.org/10.5281/zenodo.4639491

**Data availability:** The datasets used (Supplementary Table S2) are publicly available: HadCRUT5 is available at
        https://www.metoffice.gov.uk/hadobs/hadcrut5/data/current/download.html. NODC is available at
        https://www.nodc.noaa.gov/OC5/3M_HEAT_CONTENT/. Cheng et al. is available at
        http://159.226.119.60/cheng/images_files/IAP_OHC_estimate_update.txt. HadSST4 is available at
        https://www.metoffice.gov.uk/hadobs/hadsst4/data/download.html. The Global Carbon Budget 2018 data is available at
https://doi.org/10.18160/gcp-2018.

**Author contribution:** PG and BBC conceived the experiments. PG conducted the WASP model ensembles. PC and BBC analysed model output and wrote the manuscript.

**Competing interests:** The authors declare that they have no conflict of interest.

**Acknowledgements**

PG acknowledges support from UKRI Natural Environmental Research Council grant NE/T010657/1. The authors acknowledge the use of the IRIDIS High Performance Computing Facility, and associated support services at the University of Southampton, in the completion of this work. BBC acknowledges support from the National Environmental Research
Council (NE/315R015953/1) and the Horizon 2020 Framework Programme (820989, project COMFORT). The work reflects only the authors' view; the European Commission and their executive agency are not responsible for any use that may be made of the information the work contains.

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

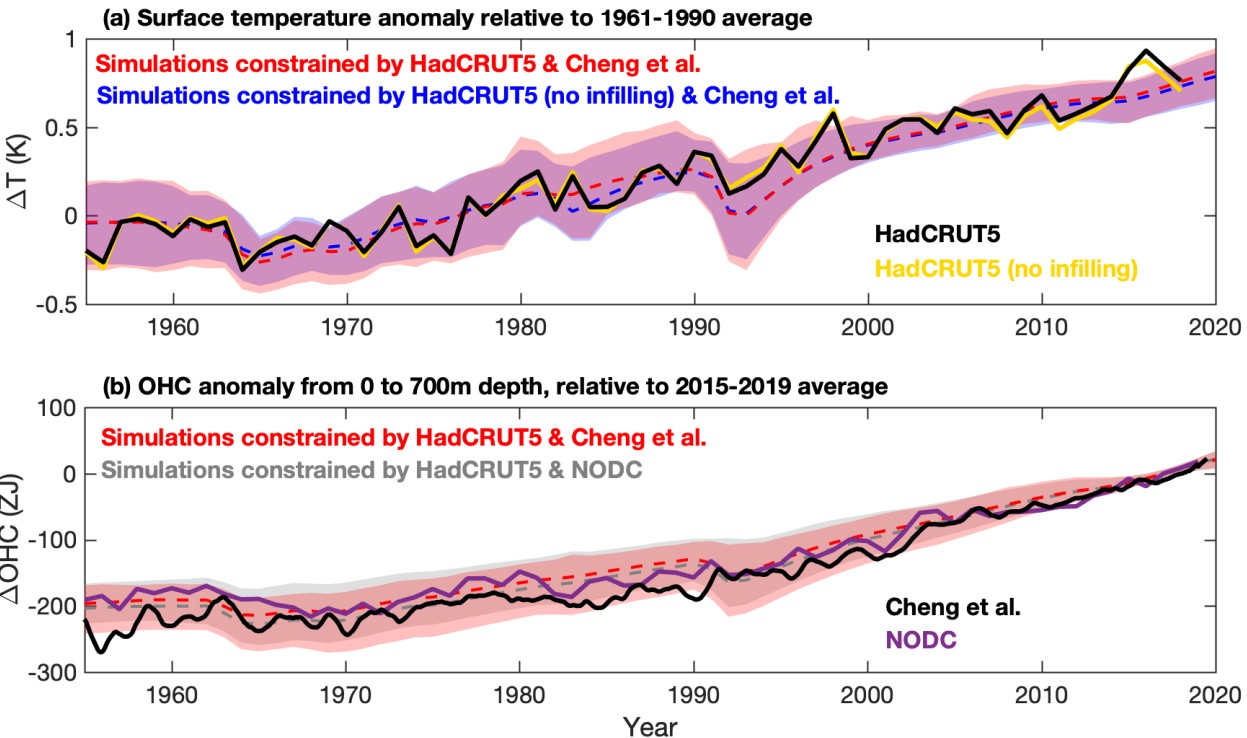

**Figure 1. Surface temperature and ocean heat content anomalies from 1955 from datasets and dataset-constrained simulations. (a)** Historic surface warming relative to the 1961-1990 average in the HadCRUT5 and HadCRUT5 (no infilling) surface temperature datasets (solid lines) and posterior ensemble simulations (dashed lines show ensemble medians, shading show 95% ensemble ranges) constrained by each temperature dataset along with the Cheng et al. ocean heat content dataset. **(b)** Ocean Heat Content (OHC) anomaly in the upper 700m of the global ocean in the NODC and Cheng et al. datasets (solid lines) and posterior ensemble simulations (dashed lines show ensemble medians, shading show 95% ensemble ranges) constrained by each OHC dataset along with the HadCRUT5 temperature dataset.

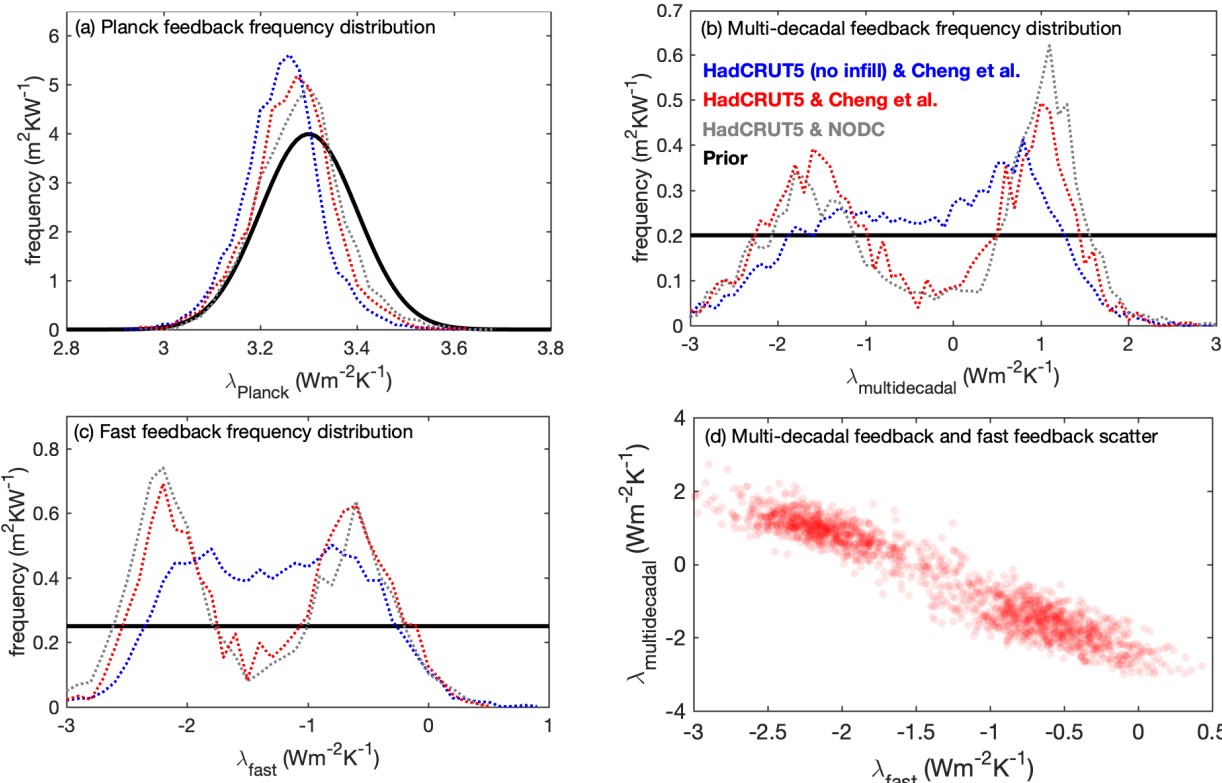

**Figure 2: Prior and posterior probability densities for climate feedback terms. (a) the Planck climate feedback; (b) fast climate feedback and (c) multi-decadal climate feedback. Shown are the prior distributions (thick black lines) and posterior distributions when constrained by different dataset combinations (dotted blue, red and grey lines). Panel (d) shows a scatter of fast climate feedback and multi-decadal climate feedback values in the posterior ensemble constrained by the HadCRUT5 and Cheng et al. datasets.**


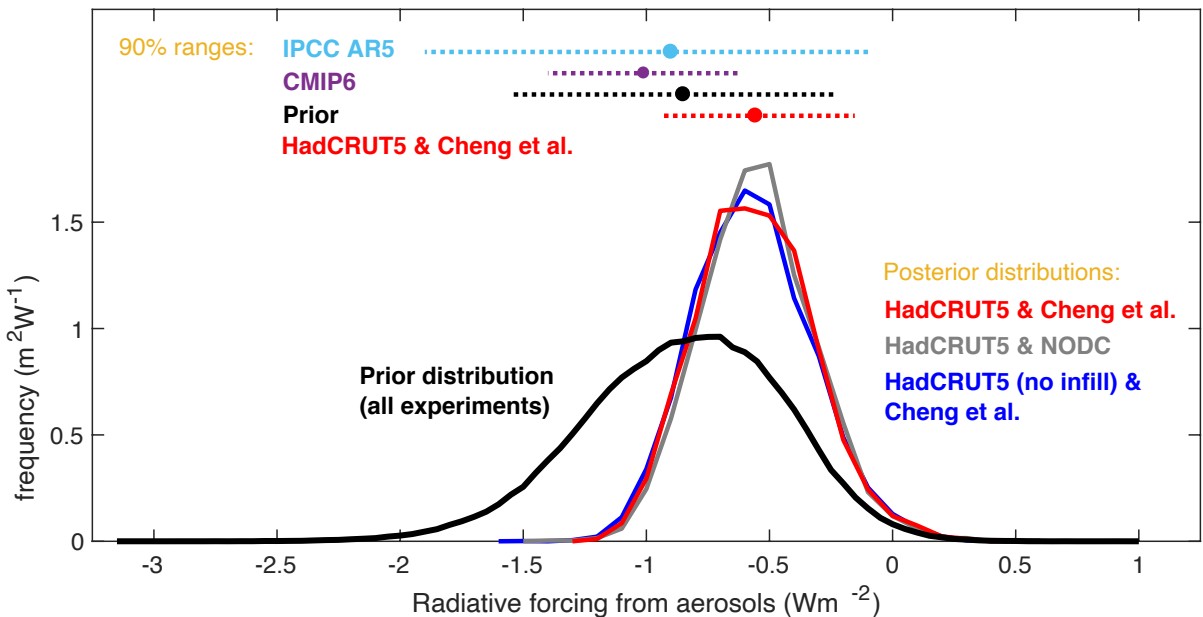

**Figure 3. Recent radiative forcing from aerosols in model ensembles and estimates. Solid lines are frequency distributions from the prior model ensembles (black) and posterior model ensembles constrained by different combinations of observational datasets (red, grey and blue). Also shown are 90% ranges (dotted lines) and best estimates (circles) from: IPCC AR5 (IPCC, 2013: light blue); an ensemble of 17 CMIP6 models analysed by Smith et al. (2020) (purple); and the prior and 'HadCRUT5 & Cheng et al.' posterior model ensembles. For model ensembles the best estimate is calculated from the model ensemble mean. The 90% range represents the 5th to 95th percentile in the Prior and HadCRUT5 & Cheng et al. model ensembles and represents the mean ±1.645 standard deviations for the CMIP6 model ensemble . All distributions are for the year 2014, expect the IPCC AR5 estimate which is for the year 2011.**



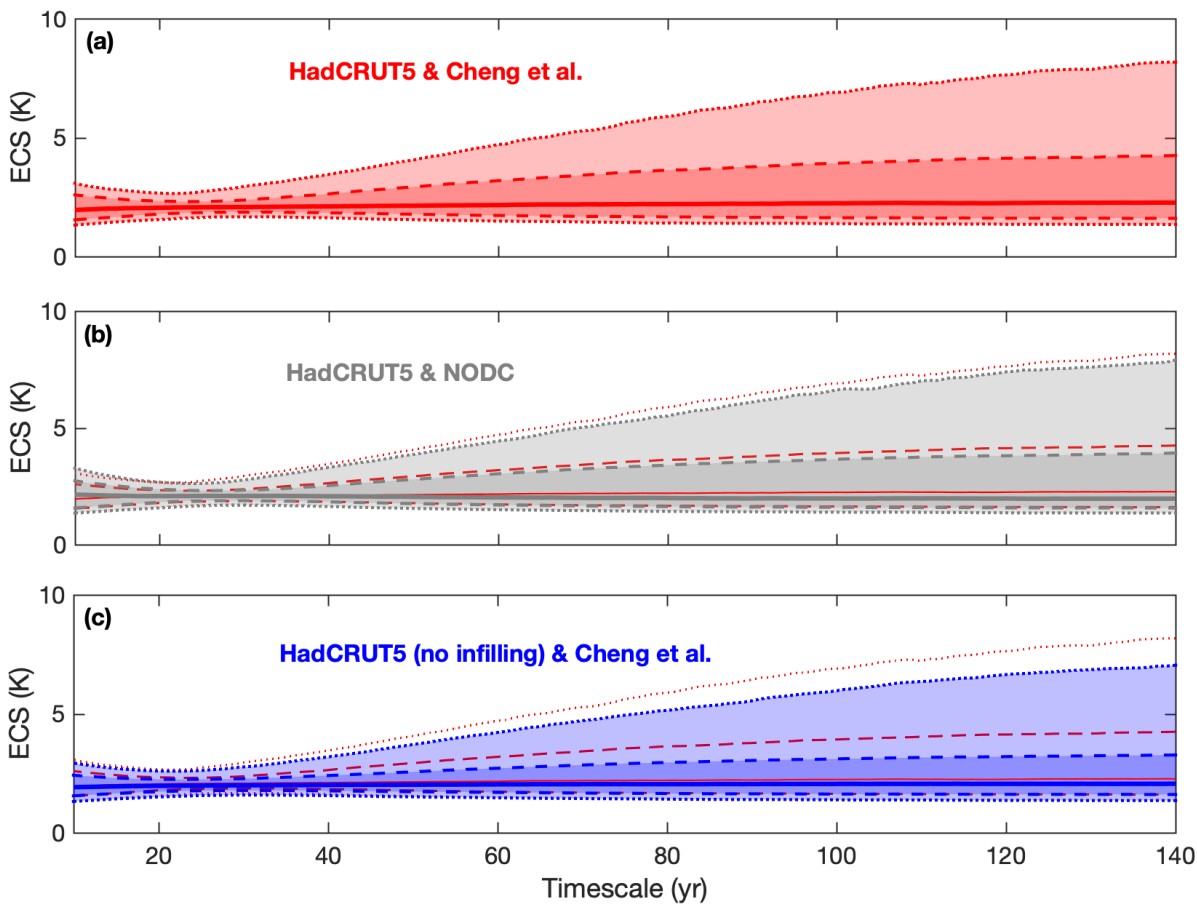

**Figure 4: Climate sensitivity (S) from 10 to 140-year response timescales following a 4xCO₂ forcing scenario constrained by different combinations of observational reconstructions. Solid lines show the median, dashed lines and dark shading show the 66% range (17th to 83rd percentiles) and dotted lines and light shading show the 95% range (2.5th to 97.5th percentiles). Panels (a), (b) and (c) show results from posterior model ensembles constrained by different dataset combinations, where red lines on panels (b) and (c) give a comparison to HadCRUT5 & Cheng et al..**


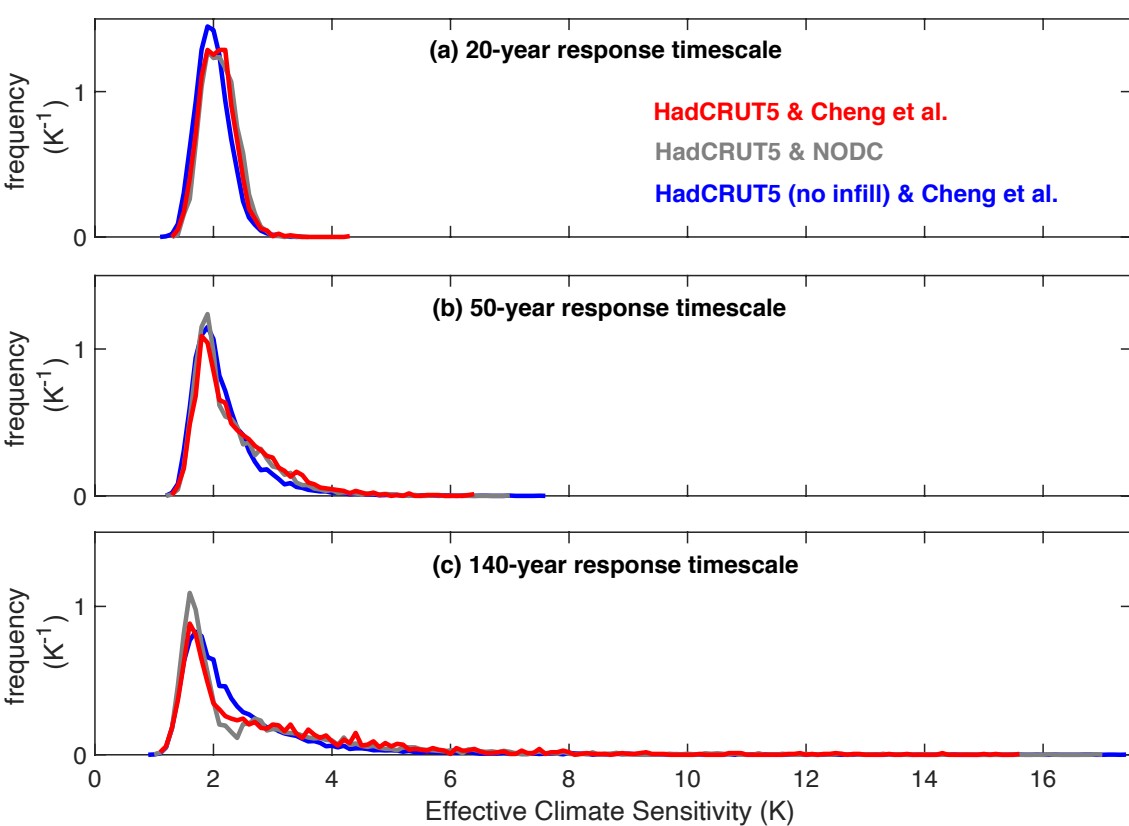

**Figure 5: Probabilistic estimates of climate sensitivity (S) for different combinations of observational constraints over (a) a 20-year response timescale, (b) a 50-year response timescale and (c) a 140-year response timescale.**

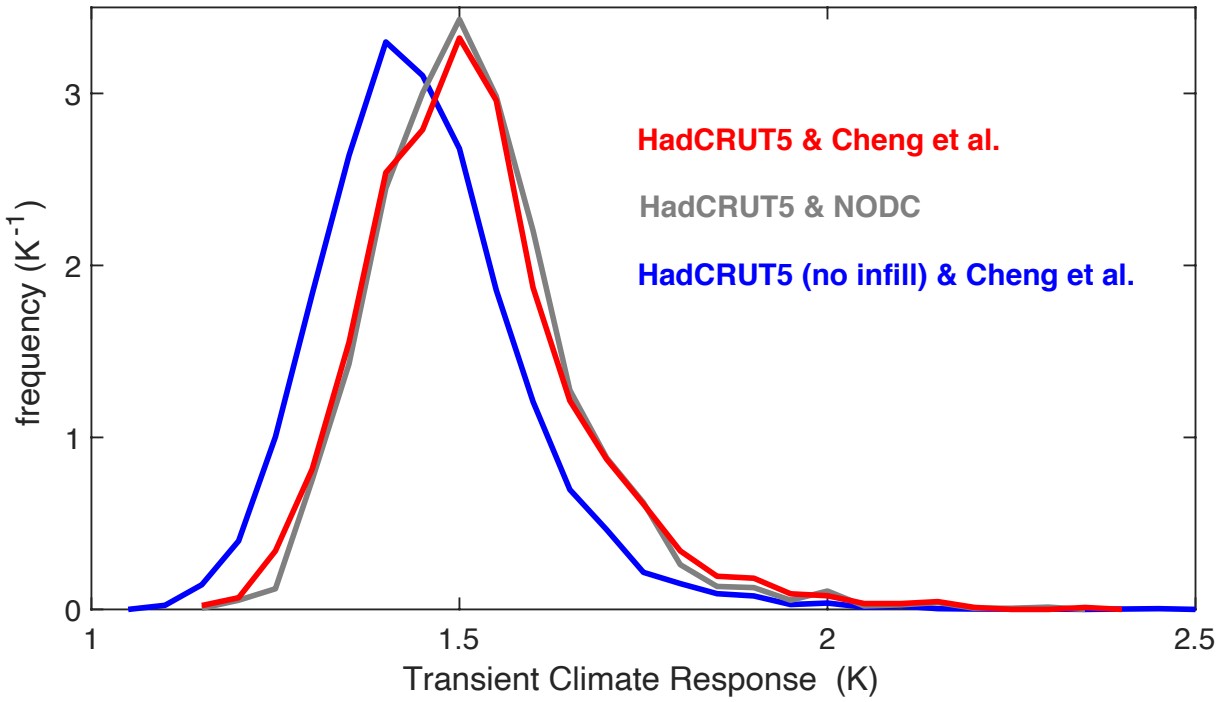

**Figure 6: Transient Climate Response (TCR) for combinations of temperature and heat content datasets, evaluated from 1pctCO2 scenario using the 20-year average warming centred on the moment of CO2 doubling.**


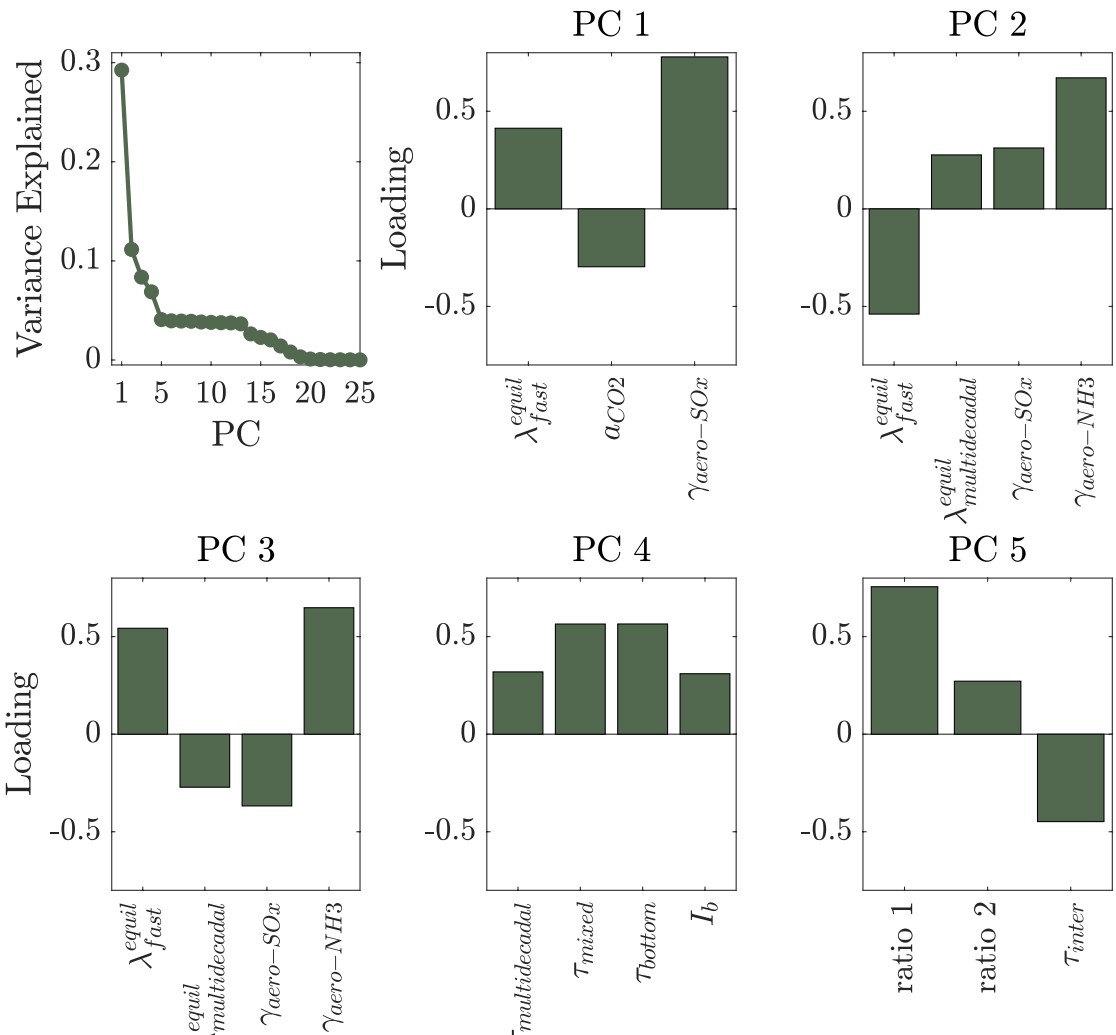

Figure 7: Principle Component Analysis of the posterior model ensembles. A) Scree plot of PC vs. variance explained. B-f) PCs 1-5 simplified by showing only the model parameters with loading greater than 0.25 in magnitude. All 25 varied model parameters are fully defined in Supplementary Table S1. Briefly: a_CO2 is the $CO_2$ radiative forcing coefficient; $\gamma_{aero-XX}$ terms reflect the sensitivity of radiative forcing to aerosol type XX; $\tau_{multidecadal}$ is the timescale for multidecadal feedback; $\tau_{mixed}$, $\tau_{inter}$ and $\tau_{bottom}$ are the ocean ventilation timescales for the ocean surface mixed layer, intermediate water and bottom water respectively, $I_b$ is the buffered carbon inventory of the air-sea system; ratio 1 is the ratio of warming for global surface temperatures relative to global sea surface temperatures at equilibrium; and ratio 2 is the ratio of warming for the whole ocean temperatures relative to sea surface temperatures at equilibrium.

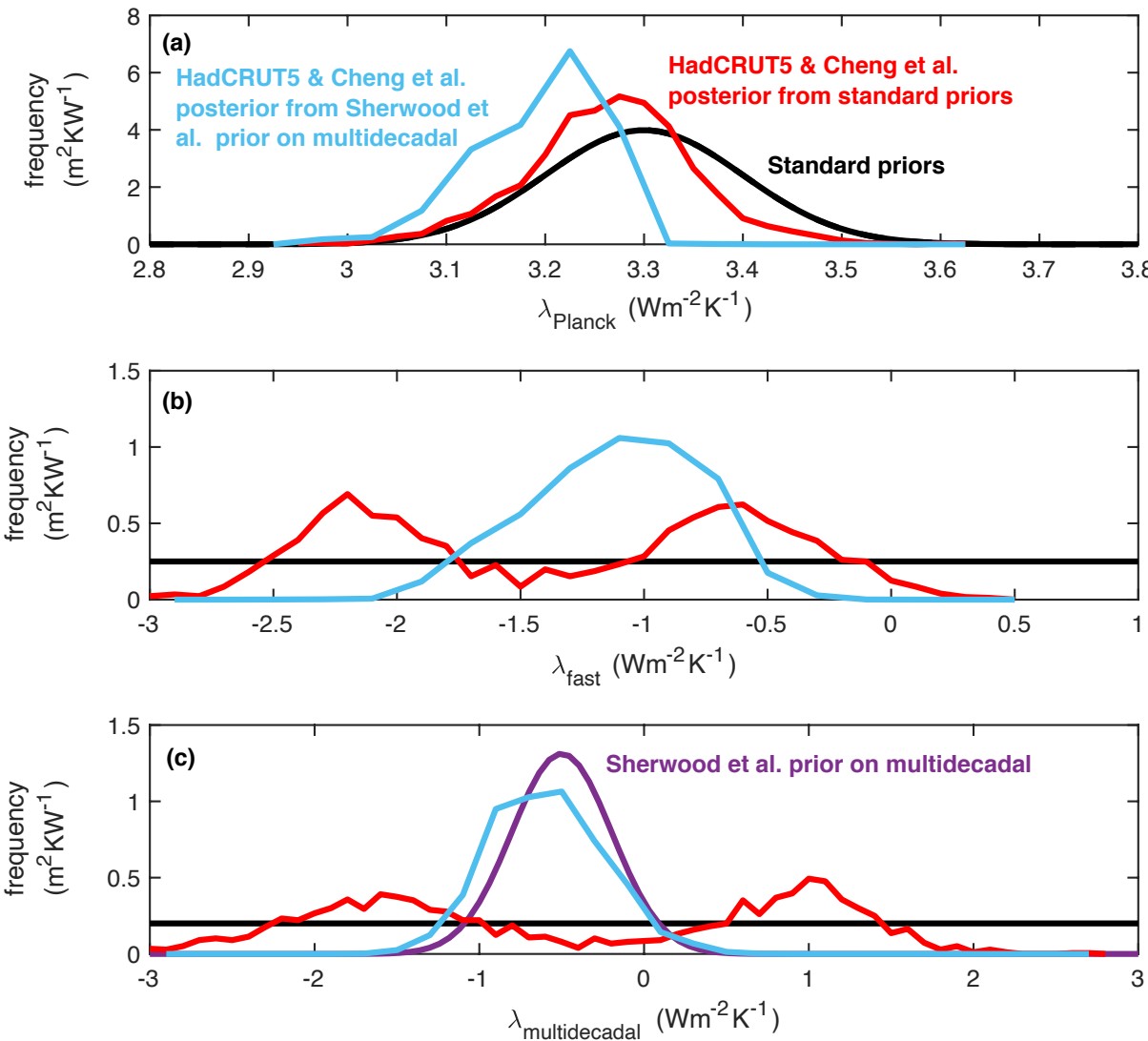

**Figure 8: The prior and posterior distribution of climate feedback terms showing the impact of alternative prior distribution for multidecadal climate feedback. Black lines show the standard prior distributions (as Fig. 2a-c) while the purple line (panel c) shows the alternative prior distribution for multidecadal feedback from Sherwood et al. (2020). Red lines show the posterior in the HadCRUT5 & Cheng et al. ensemble for the standard prior distributions (as Fig. 2a-c), light blue lines show the posterior distributions for climate feedback terms when using the alternative multidecadal prior.**

| Climate sensitivity metric | HadCRUT5 & Cheng et al. | HadCRUT5 & NODC | HadCRUT5 (no infilling) & Cheng et al. | HadCRUT5 & Cheng et al. (with alternative multidecadal climate feedback prior) |
|---|---|---|---|---|
| **S on 20-year timescale (K)** | Median: 2.1 K<br><br>66% CI: 1.8 to 2.3 K<br>90% CI: 1.6 to 2.5 K<br>95% CI: 1.6 to 2.7 K | Median: 2.1 K<br><br>66% CI: 1.8 to 2.4 K<br>90% CI: 1.7 to 2.6 K<br>95% CI: 1.6 to 2.7 K | Median: 2.0 K<br><br>66% CI: 1.7 to 2.2 K<br>90% CI: 1.6 to 2.5 K<br>95% CI: 1.5 to 2.6 K | Median: 1.9 K<br><br>66% CI: 1.7 to 2.2 K<br>90% CI: 1.6 to 2.5 K<br>95% CI: 1.6 to 2.7 K |
| **S on 50-year timescale (K)** | Median: 2.1 K<br><br>66% CI: 1.8 to 2.9 K<br>90% CI: 1.6 to 3.6 K<br>95% CI: 1.5 to 4.1 K | Median: 2.0 K<br><br>66% CI: 1.8 to 2.8 K<br>90% CI: 1.6 to 3.4 K<br>95% CI: 1.6 to 3.9 K | Median: 2.0 K<br><br>66% CI: 1.7 to 2.6 K<br>90% CI: 1.6 to 3.3 K<br>95% CI: 1.5 to 3.7 K | Median: 2.2 K<br><br>66% CI: 1.9 to 2.5 K<br>90% CI: 1.7 to 2.9 K<br>95% CI: 1.7 to 3.2 K |
| **S on 100-year timescale (K)** | Median: 2.2 K<br><br>66% CI: 1.6 to 3.9 K<br>90% CI: 1.5 to 5.7 K<br>95% CI: 1.4 to 6.9 K | Median: 2.0 K<br><br>66% CI: 1.6 to 3.7 K<br>90% CI: 1.4 to 5.4 K<br>95% CI: 1.4 to 6.6 K | Median: 2.1 K<br><br>66% CI: 1.6 to 3.1 K<br>90% CI: 1.5 to 4.8 K<br>95% CI: 1.4 to 6.0 K | Median: 2.3 K<br><br>66% CI: 2.0 to 2.7 K<br>90% CI: 1.8 to 3.2 K<br>95% CI: 1.7 to 3.6 K |
| **S on 140-year timescale (K)** | Median: 2.3 K<br><br>66% CI: 1.6 to 4.2 K<br>90% CI: 1.4 to 6.5 K<br>95% CI: 1.3 to 8.2 K | Median: 2.0 K<br><br>66% CI: 1.6 to 3.9 K<br>90% CI: 1.4 to 6.4 K<br>95% CI: 1.4 to 7.9 K | Median: 2.1 K<br><br>66% CI: 1.6 to 3.3 K<br>90% CI: 1.4 to 5.3 K<br>95% CI: 1.4 to 7.0 K | Median: 2.4 K<br><br>66% CI: 2.0 to 2.8 K<br>90% CI: 1.8 to 3.4 K<br>95% CI: 1.8 to 3.7 K |
| **TCR (K)** | Median: 1.5 K<br><br>66% CI: 1.4 to 1.6 K<br>90% CI: 1.3 to 1.8 K<br>95% CI: 1.3 to 1.9 K | Median: 1.5 K<br><br>66% CI: 1.4 to 1.6 K<br>90% CI: 1.3 to 1.8 K<br>95% CI: 1.3 to 1.8 K | Median: 1.4 K<br><br>66% CI: 1.3 to 1.6 K<br>90% CI: 1.3 to 1.7 K<br>95% CI: 1.2 to 1.8 K | Median: 1.5 K<br><br>66% CI: 1.4 to 1.6 K<br>90% CI: 1.3 to 1.7 K<br>95% CI: 1.3 to 1.8 K |

**Table 1: Climate sensitivity (S, K) and Transient Climate Response (TCR, K) best estimate (median) and ranges (where 66% Confidence Interval represents the 17th to 83rd percentile range; 90% Confidence Interval represents 5th to 95th percentile range; and 95% Confidence Interval represents 2.5th to 97.5th percentile range) under different observational constraints for surface warming and heat uptake. All ensembles use the standard prior distributions (Supplementary Table S1) except 'HadCRUT5 & Cheng et al. (with alternative multidecadal climate feedback prior)', which uses the prior for $\lambda_{multidecadal}$ from Sherwood et al. (2020) (Fig 8c, purple) and standard priors for all other terms.**