# Peer review of "Bayesian estimation of Earth's climate sensitivity and transient climate response from observational warming and heat content datasets"

_Earth System Dynamics, 2020_

## Referee Comment (RC1) · Anonymous Referee #1 · 29 Nov 2020

Summary

The authors present an update of the WASP model, using datasets up to the year 2019 of surface temperature, ocean heat content and carbon uptake. They use a time-varying feedback parameter and compare outcomes of climate response and sensitivity on different timescales and using different datasets. They complement this with an analysis of the principal components of their fitted parameters. The model are useful addition to discussions about the information it can be derived from observations and climate sensitivity, and am happy to see an updated version.

[Figure]

Major points

1. The authors compare without much comment different datasets of global warming and ocean heat uptake. The HadCRUT dataset is incomplete dataset of global temperature, with missing data at the poles, which warm faster than the average. In contrast, Cowtan&Way is an example of a dataset that does have global coverage. I would recommend switching HadCRUT out for another dataset that has taken into account polar warming (for instance NOAAGlobalTemp). Alternatively, wait (one week?) for the new version of HadCRUT, which does account for missing data.

Similarly, but probably less important, the authors compared two datasets of ocean heat uptake without comment. According to the IPCC's SROCC report, older estimates of ocean heat uptake have biases that may lead to an underestimate of ocean heat uptake (Bindoff, 2019, p.457). Cheng et al (2017) can be considered superior to the old standard of Levitus (2012).

2. I didn't get an intuitive understanding of how the time varying feedback parameter works. Why is there a difference between equation 4 and 5? It would be nice if some additional details could be included here and a reference to the first paper which you derive this.

Minor points

Abstract: it might be easier to include the 140 year response time scale, for better comparability with climate models?

L61: should multiple be two?

L 71: the first word is a typo, right?

L 83: halocarbons is not capitalised

L 92: I thought all the data used was after 1850. Why do you need volcanic aerosols before that date?

[Figure]

L111: should the j be an i?

L118: why not use the default definition of TCR of a 20 year average?

L240. This section or the discussion can do with more context. Why is this interesting? (I think it is, but I needed some brain racking!)

L344: Figs 2 -> Fig 2
* * *

---

## Referee Comment (RC2) · Anonymous Referee #2 · 30 Nov 2020

Reveiw of Goodwin and Cael for ESD.

This study uses a series of observations and a relatively simply climate model with explicit parameters to try to constrain climate sensitivity (ECS) and transient response (TCR) to CO2 doublings. The model includes feedbacks on two timescales which leads to larger ECS than what would be the case if feedback is assumed constant. Overall, I find the paper is fairly clear and fills a niche in the literature, nevertheless, I did not notice some room for improvements. Therefore I recommend only to accept this study for publications after major revisions have been undertaken.

[Figure]

Major points

I am worried that the authors are overconfident in the ability to constrain slow feedback based on historical warming. Slow feedbacks are known to evolve continuously from years to centuries (e.g. Rugenstein et al. 2020), but in this study they are limited to acting over timescales of a few decades. It is in this conjunction, where in historical warming happened for the most part over a period of ∼50 years (since the 1960-70s), that I am concerned as to whether sufficient signal is available to constrain the slow feedback. At the very least the

That said, it will probably attract attention that the authors claim to be able to constrain slow feedbacks as amplifying slow warming. Here, however, the prior assumption appears to by a uniform distribution from -3 to +2 Wm-2K-1, i.e. skewed to negative values, and thus assumed a priori to be amplifying. I would like to have the authors choose a prior that is symmetric about zero for lambda_slow.

The difference in slow feedback between the two temperature datasets is interesting. However, the explanation provided that they differ mostly be Cowtan and Way having more warming in the recent years seem insufficient. If one plots the difference over the entire record, and not just since 1960, then you realise that mostly the difference arises around the year 1900, and after 1910 the correction is remarkably stable (attached). It would seem that it should be possible to figure from where in the time series the signal that constrains slow feedback comes from?

The treatment of constraining data is also troublesome. 1) There is no particular reason to use HadCRUT without infilling. HadCRUT is only available where observations were conducted, and so has a low bias as the unobserved high latitude regions, where there is warming amplification according to climate models, are not included. Cowtan and Way infilled datasets, including that of HadCRUT but also based on other datasets such as COBE. I would suggest referring to them as 'HadCRUT in-filled', rather than
'Cowtan&Way'. 2) I am not sure why the authors include HadSST3.1 as a separate constraint, this data is already part of HadCRUT. 3) I am worried about including ocean carbon content as a constraint, atmospheric $CO_2$ is prescribed so all this does is to help constrain the exchange rates which are apparently shared with heat transfer. It is, however, well-known that the physical processes of ocean heat- and carbon uptake are different. I suggest removing this constraint.

Minor suggestions

29, Please mention here the sign convention. It seems the authors use a positive sign for the Planck feedback, which is a negative stabilising feedback, and negative signs for the positive feedbacks in the climate system (water vapor, surface albedo). Most readers will be confused over this, although I realise many British authors apply this convention.

47, Tokarska et al. (2020) only did TCR, not ECS. ECS was constrained based on recent warming by Bengtsson and Schwartz (2013), Jimenez-de-la-Cuesta and Mauritsen (2019) and Nijsse et al. (2020).

53, perhaps delete 'at any given time or timescale'

58, perhaps worthwhile mentioning those studies that are relying on these models, and why the authors of this study believe their method makes avoiding GCMs for estimating time-dependence is possible? See also major points.

71, 'Quisque' is not a word in my vocabulary. According to wikipedia it is a pre-historic herring.

81, perhaps nit-picking, but surface albedo feedback, at least that associated with sea ice, is not as fast as water vapor, see for instance Tietsche et al. (2011) that find a 1-2 year timescale.

89-90, It would be useful to display the used forcing in a figure, for example to show priors and posteriors of for example aerosol forcing, equivalent to Figure 2.

161, However, very strongly cooling aerosols would result in mid-century cooling be-
cause of the different evolutions of aerosol and greenhouse gas forcing (e.g. Stevens
2015, Bellouin et al. 2019). Supposedly the bayesian method applied automatically
filters out these values, which is why I would like to see the posterior distribution of
aerosol forcing.

204, Here, I suggest to again remind the reader of the sign convention

218, why not use a doubling of CO2? This is how ECS is defined.

224, by 90 do the authors mean 5-95?

248-250, or perhaps a better constraint on total lambda, say based on paleoclimates?

270, this section added no new information that had not already been provided. I
suggest removing it.

277, yet these components only explain 1/3 of the total variance?

353, this statement requires there are no slow feedbacks acting on timescales from
decades to millennia. I recommend to remove this statement, or strongly caveat.

368, not 'multiple' but 'two distinct' timescales.

370, IPCC 'likely' means 66 percent probability or better.

371, for Sherwood et al. (2020) probably the number referred to is 17-83 percent.
* * *
[Figure]

**Fig. 1.**

---

## Author Comment (AC1) · 10 Dec 2020

We thank both reviewers for their helpful and insightful comments. Below we show how we shall amend our manuscript for a revised submission to address the points made by Reviewer 1

Review 1

*Summary The authors present an update of the WASP model, using datasets up to the year 2019 of surface temperature, ocean heat content and carbon uptake. They use*

*a time-varying feedback parameter and compare outcomes of climate response and sensitivity on different timescales and using different datasets. They complement this with an analysis of the principal components of their fitted parameters. The model are useful addition to discussions about the information it can be derived from observations and climate sensitivity, and am happy to see an updated version.*

We thank Reviewer 1 for their careful reading of the manuscript. We agree with Reviewer1's finding that the model and method represents a useful addition to the literature, and we are pleased that the reviewer will be happy to see an updated version. Below, we specify how we will update a revised manuscript to address the points made by the reviewer.

*Major points 1. The authors compare without much comment different datasets of global warming and ocean heat uptake. The HadCRUT dataset is incomplete dataset of global temperature, with missing data at the poles, which warm faster than the average. In contrast, Cowtan and Way is an example of a dataset that does have global coverage. I would recommend switching HadCRUT out for another dataset that has taken into account polar warming (for instance NOAAGlobalTemp). Alternatively, wait (one week?) for the new version of HadCRUT, which does account for missing data. Similarly, but probably less important, the authors compared two datasets of ocean heat uptake without comment. According to the IPCC's SROCC report, older estimates of ocean heat uptake have biases that may lead to an underestimate of ocean heat uptake (Bindoff, 2019, p.457). Cheng et al (2017) can be considered superior to the old standard of Levitus (2012).*

We thank the Reviewer for highlighting the importance of the distinction between the different statistical methods are used to generate historical datasets.

Both Reviewer 1 and Reviewer 2 make clear why temperature records with infilling (e.g. Cowtan and Way) should be preferred over those without infilling (e.g. HadCRUT4). Reviewer 1 also notes that newer estimates of ocean heat uptake (e.g. Cheng et al.)

[Figure]

should be preferred over older records with identified biases.

In light of these comments from Reviewer 1 and Reviewer 2, we will highlight how our findings show that different climate sensitivities arise from these different methods of statistical historical reconstruction. Principally, a revised manuscript will highlight how HadCRUT4 with infilling (Cowtan and Way) implies a higher climate sensitivity to the standard HadCRUT4 (without infilling).

In a revised paper, we will discuss the relative merits of different data sets, and present our results in terms of the increased climate sensitivity implied when missing surface temperature anomaly data is infilled, compared to when infilling is absent. We think this is an important result, and so we would not like to remove the HadCRUT4 without infilling from our manuscript – but rather discuss the importance of infilling the surface temperature record regarding calculating climate sensitivity.

*2. I didn't get an intuitive understanding of how the time varying feedback parameter works. Why is there a difference between equation 4 and 5? It would be nice if some additional details could be included here and a reference to the first paper which you derive this.*

We agree that a revised manuscript would be improved by providing additional insight into the time varying feedback parameter within WASP. In a revised manuscript we will provide additional insight, and clearly cite the reference to the original study that presents this formulation within WASP. We will also present new figures in the supplementary material that show the time evolution of lambda according to equations (4) and (5) for idealised forcing.

Briefly, equation (4) describes how the climate feedback to an existing source of radiative forcing exponentially decays from its value at the previous time-step towards some equilibrium value. Equation 5 produces an aggregate response from: (i) the climate feedback to the existing radiative forcing (which is decayed from the value at the previous time-step towards the equilibrium value) and (ii) the climate feedback to new

radiative forcing introduced since the previous time-step (which is decayed from zero towards the new equilibrium value).

*Minor points Abstract: it might be easier to include the 140 year response time scale, for better comparability with climate models?*

We agree that improving comparability with complex climate models will enhance the manuscript, and we will consider how this should be achieved given the assumptions used in the model framework.

*L61: should multiple be two?*

Agreed that greater clarity would improve this explanation: The general code for the WASP model allows multiple climate feedbacks to act over different response timescales (see Goodwin, 2018 referenced on line 62). Here we use the Planck feedback (acting over an instantaneous timescale) plus two more feedbacks. In a revised manuscript we will state that this study chooses to consider two feedbacks, but that the WASP model may be configured for 'multiple' feedbacks.

*L 71: the first word is a typo, right?*

Agreed, this first word is a typo.

*L 83: halocarbons is not capitalised*

Agreed.

*L92: I thought all the data used was after 1850. Why do you need volcanic aerosols before that date?*

Agreed that an explanation would clarify this. The default setting for WASP model simulations is to start in the year 1765, with sources of radiative forcing defined from that date onwards. Since the temperature in year 1850 (and just afterwards) is affected by the volcanic aerosol (and all other) sources of radiative forcing just prior to 1850, we keep the with default WASP model configuration.

*L111: should the j be an i?*

Agreed, a revised manuscript will state "for each of the i sources of radiative forcing".

*L118: why not use the default definition of TCR of a 20 year average?*

Agreed, we will adopt the default definition of a 20-year average for the TCR when calculating using the WASP simulations in a revised manuscript.

*L240. This section or the discussion can do with more context. Why is this interesting?(I think it is, but I needed some brain racking!)*

Agreed that additional context on this section will improve the manuscript. This will be provided within a revised manuscript.

*L344: Figs 2 -> Fig 2*

Agreed, this will be changed.

---

## Author Comment (AC2) · 10 Dec 2020

We thank both reviewers for their helpful and insightful comments. Below we show how we shall amend our manuscript for a revised submission to address the points made by Reviewer #2.

"Review #2 This study uses a series of observations and a relatively simply climate model with explicit parameters to try to constrain climate sensitivity (ECS) and transient response (TCR) to CO2 doublings. The model includes feedbacks on two timescales

which leads to larger ECS than what would be the case if feedback is assumed constant. Overall, I find the paper is fairly clear and fills a niche in the literature, nevertheless, I did not notice some room for improvements. Therefore I recommend only to accept this study for publications after major revisions have been undertaken."

We thank Reviewer #2 for their careful reading of the manuscript and insightful comments. We are pleased the reviewer finds our manuscript clear and to fill a niche in the literature. Below, we identify how we will improve the manuscript during revision in light of the points raised by the reviewer.

"Major points I am worried that the authors are overconfident in the ability to constrain slow feedback based on historical warming. Slow feedbacks are known to evolve continuously from years to centuries (e.g. Rugenstein et al. 2020), but in this study they are limited to acting over timescales of a few decades. It is in this conjunction, where in historical warming happened for the most part over a period ofâĹij50 years (since the 1960-70s), that I am concerned as to whether sufficient signal is available to constrain the slow feedback. At the very least the"

We agree that the method used in the manuscript does not constrain the 'slow feedback' consisting of all feedbacks evolving over timescales from years to centuries. On re-reading the manuscript in light of the reviewer's comments, we agree that the manuscript could appear overconfident in its ability to provide a constraint on this 'slow feedback'. A revised manuscript will introduce clarity in this area, carefully caveating the methodology and results.

The following changes will be employed:

(1) We will re-name lambda_slow to reflect the timescales that it does (and does not) consider:

What our manuscript has named 'lambda_slow' is not in fact the total slow feedback evolving from years to centuries, and we see how this naming of lambda_slow may

cause confusion for readers into thinking that it does represent all slow feedbacks.

In fact, 'lambda_slow' in our mansucript represents only the feedbacks acting with multi-decadal ($\sim$ 25 to 40 year) e-folding timescales. One key slow feedback acting on such timescales identified within the literature is often termed the pattern effect, whereby sea surface warming patterns evolve over a multi-decadal timescale and change lambda. However, since our method cannot separate this pattern effect from contributions of any other feedbacks acting on similar timescales, we adopted the term lambda_slow as a catch-all. The problem with lambda_slow (as identified by the reviewer) is that it implies we are considering slow climate feedbacks acting on longer timescales as well (which the reviewer rightly points out that our method does not consider).

In a revised manuscript, we will adopt a different name for lambda_slow that explicitly reflects the multi-decadal timescales that it considers. We will also explicitly state that our multi-decadal feedback term does not include other slow feedbacks that evolve over longer (e.g. century) timescales.

What our methodology has achieved is to explore possible combinations of fast climate feedback and multi-decadal feedback that are consistent with historical observations of global mean temperature and heat content anomalies. We agree that the historical record is insufficiently long to fully explore longer timescale (century) feedbacks, and we have not done so.

(2) We identify regions of parameter space with both amplifying and damping multi-decadal feedbacks that are consistent with observations. We will highlight that our results are perfectly compatible with multi-decadal feedbacks that act to dampen the increase in global temperatures – even though our best estimate is for multi-decadal feedbacks that amplify warming.

"That said, it will probably attract attention that the authors claim to be able to constrain slow feedbacks as amplifying slow warming. Here, however, the prior assumption appears to by a uniform distribution from -3 to +2 Wm-2K-1, i.e. skewed to negative values, and thus assumed a priori to be amplifying. I would like to have the authors choose a prior that is symmetric about zero for lambda_slow."

Agreed that it is important to show that the best estimate of amplifying multi-decadal feedback in our results is not a result of the prior distribution we have adopted. In a revised manuscript, as suggested, we will extend our exploration of parameter space by also considering the section with multi-decadal feedback from +2 Wm-2K-1 to +3Wm-2K-1. This will then have sampled parameter space from -3Wm-2K-1 to +3Wm-2K-1, and so will not be a prior skewed to amplifying values.

Note that the posterior distribution for lambda_slow (the culti-decadal feedback) is already firmly in the tail of the distribution before getting to +2 Wm-2K-1 (Figure 2, dotted and dashed red lines), and so we anticipate relatively small numbers of posterior simulations to be identified beyond +2Wm-2K-1.

"The difference in slow feedback between the two temperature datasets is interesting. However, the explanation provided that they differ mostly be Cowtan and Way having more warming in the recent years seem insufficient. If one plots the difference over the entire record, and not just since 1960, then you realise that mostly the difference arises around the year 1900, and after 1910 the correction is remarkably stable (attached). It would seem that it should be possible to figure from where in the time series the signal that constrains slow feedback comes from?"

We thank the reviewer for pointing this out, including with their useful figure. We Agree that the difference in temperature records is not simply that the Cowtan and Way record has more warming since 1960, and that it is more complicated. In a revised version we will amend the manuscript to reflect this. We do not think that our methodology (so far) will allow us to state with clarity where in the temperature records the key differences are that affect the constraint on the slower feedback. We agree that this is interesting and will reserve this for future work.

"The treatment of constraining data is also troublesome. 1) There is no particular reason to use HadCRUT without infilling. HadCRUT is only available where observations wereconducted, and so has a low bias as the unobserved high latitude regions, where thereis warming amplification according to climate models, are not included. Cowtan andWay infilled datasets, including that of HadCRUT but also based on other datasetssuch as COBE. I would suggest referring to them as 'HadCRUT in-filled', rather than 'Cowtan and Way'"

We include HadCRUT4 without infilling as it contains more sources of uncertainty in the published uncertainty estimates. We use Cowtan and Way (i.e. 'HadCRUT with infilling') because it statistically infills missing regions of data. In our original manuscript we avoided making judgements on the relative merits of different datasets. Reviewers #1 and #2 have both highlighted valid reasons for preferring particular datasets due to their methodologies. In a revised manuscript we will also highlight these reasons for preferring the estimates of climate sensitivity from infilled records of temperature anomaly.

"2) I am not sure why the authors include HadSST3.1 as a separate constraint, this data is already part of HadCRUT."

We agree that the reasons for including a sea surface temperature constraint in addition to global mean surface temperature should be explained in a revised manuscript. Briefly: The WASP model contains an input parameter stating the ratio of global sea-surface warming to global mean surface warming at equilibrium ($r1$ noted in the supplementary material). As this parameter varies between ensemble members, simulated global mean surface warming and sea surface temperature warming may vary differently (relative to each other) across the ensemble. Therefore, one constraint for 'SST warming only' is required to help constrain the posterior values of $r1$ within the WASP ensembles, hence the use of HadSST3.

"3) I am worried about including ocean carbon content as a constraint, atmospheric

$CO_2$ is prescribed so all this does is to help constrain the exchange rates which are apparently shared with heat transfer. It is, however, well-known that the physical processes of ocean heat- and carbon uptake are different. I suggest removing this constraint."

We agree that the use of ocean carbon uptake should be explained in a revised manuscript, and we agree that the processes of heat and carbon uptake by the ocean are different.

The WASP model contains a specific parameter that speficies how different the processes of heat and carbon uptake by the ocean are in the simulation ($r2$, noted in the supplementary material). This parameter ($r2$) is varied between ensemble members, and the only way to constrain the values of $r2$ that reach the posterior distribution is to include ocean carbon uptake (alongside ocean heat uptake) as one of the historic constraints. If we did not use ocean carbon uptake as one of the historic constraints, then the WASP model could achieve acceptable ocean heat uptake levels with unrealistic input parameter values for ocean circulation timescales alongside a compensating unrealistic value for $r2$ – hence the use of ocean carbon uptake alongside ocean heat uptake constraints. We will explain this in a revised manuscript.

"Minor suggestions

29, Please mention here the sign convention. It seems the authors use a positive sign for the Planck feedback, which is a negative stabilising feedback, and negative signs for the positive feedbacks in the climate system (water vapor, surface albedo). Most readers will be confused over this, although I realise many British authors apply this convention."

Agreed that there are two sign conventions in use in the literature for climate feedback. We will explain in a revised manuscript that our sign convention derives from the definition of lambda as the 'increase in outgoing radiation for a 1K rise in global mean surface temperature'. We will also highlight what this means for amplifying and

damping feedback processes in terms of the sign of lambda.

"47, Tokarska et al. (2020) only did TCR, not ECS. ECS was constrained based on recent warming by Bengtsson and Schwartz (2013), Jimenez-de-la-Cuesta and Mauritsen (2019) and Nijsse et al. (2020)."

We thank the reviewer for these recommendations, we will amend the references cited here in a revised version.

"53, perhaps delete 'at any given time or timescale'"

Agreed, this will be deleted in a revised manuscript.

"58, perhaps worthwhile mentioning those studies that are relying on these models, and why the authors of this study believe their method makes avoiding GCMs for estimating time-dependence is possible? See also major points."

Agreed that citing literature using GCMs to estimate time-evolving climate sensitivity here would improve the manuscript for the reader. In a revised manuscript, we will also explain here how our methodology works: We sample values of fast climate feedback and multi-decadal climate feedback (and other parameters) looking for combinations that give rise to historic warming and heat content anomalies that are consistent with observations.

"71, 'Quisque' is not a word in my vocabulary. According to wikipedia it is a pre-historic herring."

Thank you, this word was a typo and shall not appear in a revised version.

"81, perhaps nit-picking, but surface albedo feedback, at least that associated with seaice, is not as fast as water vapor, see for instance Tietsche et al. (2011) that find a 1-2year timescale."

Agreed, it is true that the surface sea-ice albedo component of the fast feedback does strictly have a timescale longer than the residence timescale of water vapour in the

atmosphere. We shall mention this in a revised manuscript.

"89-90, It would be useful to display the used forcing in a figure, for example to show priors and posteriors of for example aerosol forcing, equivalent to Figure 2."

Thank you for highlighting this, we agree this would be beneficial. Yes - in a revised manuscript we will display a range of forcing figures, either in the main text or supplementary material, showing the priors and posteriors for different sources of radiative forcing (including aerosols).

"161, However, very strongly cooling aerosols would result in mid-century cooling because of the different evolutions of aerosol and greenhouse gas forcing (e.g. Stevens2015, Bellouin et al. 2019). Supposedly the bayesian method applied automatically filters out these values, which is why I would like to see the posterior distribution of aerosol forcing."

We agree that very cooling aerosols would result in mid-century cooling. In a revised manuscript, we will present the posterior distribution of aerosol forcing as a new figure in the main text or supplementary material. We also agree that our Bayesian approach filters out combinations of aerosol and greenhouse gas forcing that result in unrealistic evolutions of historic temperature or heat content anomalies.

"204, Here, I suggest to again remind the reader of the sign convention"

Agreed, in a reivsed manuscript we shall remind the reader of the sign convention adopted for climate feedback again here.

"218, why not use a doubling of CO2? This is how ECS is defined."

We agree that a CO2 doubling would also work here. We have chosen to use a 4xCO2 perturbation, in line with on of the standard idealised scenarios for CMIP-class models. Note that WASP does not (yet) have a state-dependence on lambda and so the results of a 2xCO2 experiment would be equivalent (but with a slightly lower signal to noise ratio, where the noise is driven by the imposed interannual variability in Earth energy

balance in WASP: see line 96).

"224, by 90 do the authors mean 5-95?"

Agreed that this was unclear, in a revised manuscript we will specify the upper and lower percentile bounds of confidence intervals as well as the sizes of the intervals.

"248-250, or perhaps a better constraint on total lambda, say based on paleoclimates?"

Agreed, we will mention that other approaches (such a palaeoclimate) would help with constraining total climate feedback.

"270, this section added no new information that had not already been provided. I suggest removing it. 277, yet these components only explain 1/3 of the total variance?"

We agree our manuscript that lacked clarity on the benefits and motivation behind the principle component and stepwise regression sections (4.2.2 and 4.2.3 respectively). We will explain the motivation and benefits of these sections more clearly in a revised manuscript. Briefly: it is advantageous to understand how many degrees of freedom a climate model has for being observation-consistent with global temperature and heat content constraints up to the present day. The fact that a large fraction of the variance (around 1/3) in the posterior ensemble is explained by a much smaller number of degrees of freedom than exists in the prior model ensemble is a significant finding. In part, understanding the posterior ensemble's degrees of freedom may help to sample parameter space with a smaller number of ensemble members – which becomes more significant for constructing ensembles with complex model that are more computationally expensive.

"353, this statement requires there are no slow feedbacks acting on timescales from decades to millennia. I recommend to remove this statement, or strongly caveat."

Agreed, there are many slow feedbacks acting on many timescales from multi-decadal to millennial. What our manuscript explores is a particular multi-decadal feedback-timescale, rather than all slow feedback timescales. In a revised manuscript we will rename lambda_slow to (e.g. lambda_md, for lambda_multi-decadal). This will remove confusion (both in this section and elsewhere) about which slow feedback timescales we explore, and which we do not explore, throughout the manuscript.

"368, not 'multiple' but 'two distinct' timescales."

Agreed that greater clarification is required. The WASP model code allows for 'multiple' timescales to be considered, but we will specify in revision that in this manuscript we have considered precisely two distinct timescales (in addition to the instantaneous Planck feedback).

"370, IPCC 'likely' means 66 percent probability or better."

Agreed, we will clarify the sentence in a revised manuscript to specify IPCC definition of 'likely'.

"371, for Sherwood et al. (2020) probably the number referred to is 17-83 percent."

Agreed, we will clarify this in a revised manuscript.

---

## Author Response (AR1)

We thank both reviewers for their helpful and insightful comments. Below we show how we have amend our manuscript in this revised submission to address the points made, first by Reviewer #1 and then by Reviewer #2.

Review #1

*Summary*
*The authors present an update of the WASP model, using datasets up to the year 2019 of surface temperature, ocean heat content and carbon uptake. They use a time-varying feedback parameter and compare outcomes of climate response and sensitivity on different timescales and using different datasets. They complement this with an analysis of the principal components of their fitted parameters. The model are useful addition to discussions about the information it can be derived from observations and climate sensitivity, and am happy to see an updated version.*

We thank Reviewer #1 for their careful reading of the manuscript. We agree with Reviewer#1's finding that the model and method represents a useful addition to the literature, and we are pleased that the reviewer will be happy to see an updated version. Below, we specify how we have updated our revised manuscript to address the points made by the reviewer.

*Major points*
*1. The authors compare without much comment different datasets of global warming and ocean heat uptake. The HadCRUT dataset is incomplete dataset of global temperature, with missing data at the poles, which warm faster than the average. In contrast, Cowtan and Way is an example of a dataset that does have global coverage. I would recommend switching HadCRUT out for another dataset that has taken into account polar warming (for instance NOAAGlobalTemp). Alternatively, wait (one week?) for the new version of HadCRUT, which does account for missing data. Similarly, but probably less important, the authors compared two datasets of ocean heat uptake without comment. According to the IPCC's SROCC report, older estimates of ocean heat uptake have biases that may lead to an underestimate of ocean heat uptake (Bindoff, 2019, p.457). Cheng et al (2017) can be considered superior to the old standard of Levitus (2012).}*

We thank the Reviewer for highlighting the importance of the distinction between the different statistical methods are used to generate historical datasets.

Both Reviewer #1 and Reviewer #2 make clear why temperature records with infilling should be preferred over those without infilling. Reviewer #1 also notes that newer estimates of ocean heat uptake (e.g. Cheng et al.) should be preferred over older records with identified biases.

In light of these comments from Reviewer #1 and Reviewer #2, we have highlighted how our findings show that different climate sensitivities arise from these different methods of statistical historical reconstruction. Principally, this revised manuscript has updated to the new HadCRUT5 dataset, which includes statistical infilling, and compares to the HadCRUT5 without statistical infilling dataset. We clearly set out how the HadCRUT5 dataset is our preferred choice for constraining climate sensitivity (Lines 205-210):

"The preferred combination of observational datasets is HadCRUT5 & Cheng et al., as these represent the most up to date methodologies for their respective temperature (Morice et al., 2021) and heat content (Cheng et al., 2017) reconstructions. The other dataset combinations are included to assess the sensitivity of our method to different heat content datasets (HadCRUT5 & NODC) and the sensitivity of our findings to the statistical infilling of missing data (HadCRUT5 (no infill) & Cheng et al.). It is noted that most other temperature datasets now reconstruct similar historic global mean temperature anomalies to HadCRUT5 (e.g. see Morice et al. 2021)."

*2. I didn't get an intuitive understanding of how the time varying feedback parameter works. Why is there a difference between equation 4 and 5? It would be nice if some additional details could be included here and a reference to the first paper which you derive this.*

The revised manuscript now contains an extensive explanation of the representation of time-varying climate feedbacks in the supplementary material (Supplementary Information Section S3), including a new Supplementary Figure S7 giving an example of climate feedback responses to two idealised step-function scenarios in radiative forcing.

*Minor points*
*Abstract: it might be easier to include the 140 year response time scale, for better comparability with climate models?*

We agree that improving comparability with complex climate models will enhance the manuscript, and we have presented 140-year response timescale analysis to be directly comparable with complex model experiments run for this length of time (see revised version Figures 4, 5 and Table 1). The figures quoted in the abstract refer to the 140-year ECS values we calculate to

*L61: should multiple be two?*

Agreed that greater clarity is required: The general code for the WASP model allows multiple climate feedbacks to act over different response timescales (see Goodwin, 2018 referenced on line 62). Here, we use the Planck feedback (acting over an instantaneous timescale) plus two more feedbacks. Our revised manuscript now states this (Lines 61-64):

"This study considers the instantaneous Planck feedback and two further timescales of climate feedback: a multi-diurnal feedback representing a selection of fast climate processes, such as water vapour and clouds, and a multi-decadal climate feedback representing slower processes, such as the surface warming pattern effect."

*L 71: the first word is a typo, right?}*

Agreed, this first word is a typo and has been removed.

*L 83: halocarbons is not capitalised}*

Agreed, halocarbons now a appear uncapitalized throughout (Line 98 of main manuscript, and also see Supplementary Information).

*L92: I thought all the data used was after 1850. Why do you need volcanic aerosols before that date?}*

Agreed that an explanation helps clarify this. The default setting for WASP model simulations is to start in the year 1765 for RCP scenarios or 1700 for ssp scenarios, with sources of radiative forcing defined with non-zero values from some date after the model start date. Since the temperature in year 1850 (and just afterwards) is affected by the volcanic aerosol (and all other) sources of radiative forcing just prior to 1850, we keep the with default WASP model configuration.

This is now explained (Lines 92-95):

"The WASP model starts simulations at year 1700 by default (e.g. Goodwin, 2018), with different sources of radiative forcing defined from some time after that date. While the observational constraints used in this study start in year 1850, the model state in 1850 is affected by radiative forcing received prior to that date. Therefore, this study imposes radiative forcing on the WASP model prior to 1850."

*L111: should the j be an i?}*

Agreed, the sentence now states (Line 124-125):

"for each of the *i* sources of radiative forcing".

*L118: why not use the default definition of TCR of a 20 year average?*

Agreed, we have adopted the default definition of a 20-year average for the TCR when calculating using the WASP simulations in a revised manuscript. (e.g. Lines 130-132):

"Here, the transient climate response, TCR, is calculated as the 20-year average warming centred at the year of $CO_2$ doubling for a scenario with a 1 per cent per year rise in $CO_2$ and no other forcing (hereafter: 1pct$CO_2$ scenario)."

*L240. This section or the discussion can do with more context. Why is this interesting?(I think it is, but I needed some brain racking!)*

Agreed. The key point are that this type of analysis may ultimately help reduce uncertainty in the ECS and TCR by simplifying what the sources of uncertainty are. Also, conducting this type of statistical analysis on efficient model ensembles may ultimately help analyse complex model ensembles. The additional context is now provided for this section on Lines 272-277:

"The observational records provide constraints on the parameters of the posterior ensembles that manifest not only as posterior distributions for these parameters but also as relationships between them, as well as between model parameters and key model outputs of interest (such as ECS(t)). While the correlation structure of the 25 parameters' joint posterior distribution is generally quite complex, some key structures emerge that indicate how ECS and TCR uncertainties might be reduced. This method of analysing variation, and simplifying the degrees of freedom of variation, in large data-constrained efficient model ensembles may ultimately help explore parameter space in more complex Earth system models."

*L344: Figs 2 -> Fig 2*

Agreed, this has been changed.

*Review #2 This study uses a series of observations and a relatively simply climate model with explicit parameters to try to constrain climate sensitivity (ECS) and transient response (TCR) to CO2 doublings. The model includes feedbacks on two timescales which leads to larger ECS than what would be the case if feedback is assumed constant. Overall, I find the paper is fairly clear and fills a niche in the literature, nevertheless, I did not notice some room for improvements. Therefore I recommend only to accept this study for publications after major revisions have been undertaken.*

We thank Reviewer #2 for their careful reading of the manuscript and insightful comments. We are pleased the reviewer finds our manuscript clear and to fill a niche in the literature. Below, we identify how have improved the manuscript during revision in light of the points raised by the reviewer.

*Major points*
*I am worried that the authors are overconfident in the ability to constrain slow feedback based on historical warming. Slow feedbacks are known to evolve continuously from years to centuries (e.g. Rugenstein et al. 2020), but in this study they are limited to acting over timescales of a few decades. It is in this conjunction, where in historical warming happened for the most part over a*

*period of ~50 years (since the 1960-70s), that I am concerned as to whether sufficient signal is available to constrain the slow feedback. At the very least the*

We agree that the method used in the manuscript does not constrain the 'slow feedback' consisting of all feedbacks evolving over timescales from years to centuries. Rather, our manuscript uses the historical record to get a constraint on the multidecadal feedback (timescale 25-40 years). To avoid confusion, we have amended the terminology used in the manuscript: the term that was called lambda_slow in the previous version is, in this revised version, now renamed lambda_multidecadal (e.g. lines 83, 123, 163, 221, …)

Also, when lambda_multidecadal is introduced we now explicitly state that this manuscript does not attempt to constrain feedbacks acting over timescales slower than multidecadal (Lines 86-88):

> "Note that slow climate feedbacks with timescales longer than multi-decadal are not explored here, since the historical records of temperature and heat content changes do not extend long enough to offer a reliable constraint on processes acting on such long timescales."

*"That said, it will probably attract attention that the authors claim to be able to constrain slow feedbacks as amplifying slow warming. Here, however, the prior assumption appears to by a uniform distribution from -3 to +2 Wm-2K-1, i.e. skewed to negative values, and thus assumed a priori to be amplifying. I would like to have the authors choose a prior that is symmetric about zero for lambda_slow."*

Agreed, this revised manuscript now uses a prior for lambda_multidecadal that is centred on zero, and so does not assume a priori that lambda_multidecadal is either amplifying or damping. Our new prior for lambda_multidecadal is a uniform distribution from -3Wm-2K-1 to +3Wm-2K-1, thus with an equal chance of being damping as being amplifying (See Figure 2b and Supplementary Table S1).

Also, the revised manuscript now explicitly states how there are regions of our posterior parameter space with amplifying lambda_multidecadal and there are also regions of our posterior parameter space with damping lambda_multidecadal, so it is clear to the reader that our revised manuscript is consistent with both amplifying or damping multidecadal climate feedbacks (Lines 242-247):

> "The posterior distributions for fast and multi-decadal climate feedback strengths are bimodal in the HadCRUT5 & Cheng et al. and HadCRUT5 & NODC ensembles (Fig. b,c, red and grey), corresponding to one observation consistent region with weaker amplifying fast feedback ($\lambda_{fast}^{equil} \sim -0.6$ Wm$^{-2}$) and strong amplifying multidecadal feedback ($\lambda_{multidecadal}^{equil} \sim -1.7$Wm$^{-2}$) , and another observation consistent region with very strong amplifying fast feedback ($\lambda_{fast}^{equil} \sim -2.2$Wm$^{-2}$) and damping multidecadal feedback ($\lambda_{multidecadal}^{equil} \sim +1$Wm$^{-2}$) (Fig. 2d, shown for the HadCRUT5 & Cheng et al. ensemble), noting that the sign convention used implies amplifying feedback from negative $\lambda$."

*The difference in slow feedback between the two temperature datasets is interesting. However, the explanation provided that they differ mostly be Cowtan and Way having more warming in the recent years seem insufficient. If one plots the difference over the entire record, and not just since 1960, then you realise that mostly the difference arises around the year 1900, and after 1910 the correction is remarkably stable (attached). It would seem that it should be possible to figure from where in the time series the signal that constrains slow feedback comes from?*

Agreed, with the revised datasets (HadCRUT5 replacing HadCRUT4) this finding is removed from the study.

*The treatment of constraining data is also troublesome. 1) There is no particular reason to use HadCRUT without infilling. HadCRUT is only available where observations were conducted, and so has a low bias as the unobserved high latitude regions, where there is warming amplification according to climate models, are not included. Cowtan and Way infilled datasets, including that of HadCRUT but also based on other datasets such as COBE. I would suggest referring to them as 'HadCRUT in-filled', rather than 'Cowtan and Way'*

Reviewers #1 and #2 have both highlighted valid reasons for preferring particular datasets due to their methodologies. In a revised manuscript we will also highlight these reasons for preferring the estimates of climate sensitivity from infilled records of temperature anomaly.

*2) I am not sure why the authors include HadSST3.1 as a separate constraint, this data is already part of HadCRUT.*

We now explain the reasons why HadSST4 (in the new revised version) is used as an additional constraint to HadCRUT5 in the Supplementary Information, Supplementary Section S4:

"The WASP model contains one input parameter for the ratio of global sea-surface warming to global mean surface warming at equilibrium (Ratio 1 or r1 in Supplementary Table S1) and another for the ratio of global whole-ocean warming to global sea-surface warming at equilibrium (Ratio 2 or r2 in Supplementary Table S1). It is these input parameters that require the use of a separate observational constraint for sea surface temperatures (HadSST4 in Supplementary Table S2) and an observational constraint for ocean carbon uptake (The Global Carbon Budget in Supplementary Table S2) to be applied. As the r1 parameter is varied between prior ensemble members, simulated global mean surface warming and sea surface temperature warming vary differently (relative to each other) across the ensemble. Therefore, the observational constraints for both global surface temperature and sea surface temperature are required to help constrain the posterior values of r1 within the posterior ensembles. As the r2 parameter is varied between prior ensemble members, the relative ocean uptakes of heat and carbon are varied across the ensemble members. Therefore, observational constraints for both ocean heat and carbon uptake are required to constrain the values of r2 in the posterior ensembles."

*3) I am worried about including ocean carbon content as a constraint, atmospheric CO2 is prescribed so all this does is to help constrain the exchange rates which are apparently shared with heat transfer. It is, however, well-known that the physical processes of ocean heat- and carbon uptake are different. I suggest removing this constraint.*

We now explain the use of separate ocean heat and carbon uptake constraints to generate the posterior ensembles (Supplementary Section S4):

""The WASP model contains one input parameter for the ratio of global sea-surface warming to global mean surface warming at equilibrium (Ratio 1 or r1 in Supplementary Table S1) and another for the ratio of global whole-ocean warming to global sea-surface warming at equilibrium (Ratio 2 or r2 in Supplementary Table S1). It is these input parameters that require the use of a separate observational constraint for sea surface temperatures (HadSST4 in Supplementary Table S2) and an observational constraint for ocean carbon uptake (The Global Carbon Budget in Supplementary Table S2) to be applied. As the r1 parameter is varied between prior ensemble members, simulated global mean surface warming and sea surface temperature warming vary differently (relative to each other) across the ensemble. Therefore, the observational constraints for both global surface temperature and sea surface temperature are required to help constrain the posterior values of r1 within the posterior ensembles. As the r2 parameter is varied between prior ensemble members, the relative ocean uptakes of heat and carbon are varied across the ensemble members. Therefore, observational constraints for both ocean heat and carbon uptake are required to constrain the values of r2 in the posterior ensembles."

*Minor suggestions*

*29, Please mention here the sign convention. It seems the authors use a positive sign for the Planck feedback, which is a negative stabilising feedback, and negative signs for the positive feedbacks in the climate system (water vapor, surface albedo). Most readers will be confused over this, although I realise many British authors apply this convention.*

Agreed that there are two sign conventions in use in the literature for climate feedback. We adopt the sign convention of positive lambda is physically meaningful. We now initially state this sign convention on Line 91:

"The sign convention adopted has positive overall $\lambda_{eff}$, such that negative $\lambda_{fast}$ and $\lambda_{multidecadal}$ are amplifying."

and mention again where it impacts interpreting the results on line 248:

"… noting that the sign convention used implies amplifying feedback from negative $\lambda$."

*47, Tokarska et al. (2020) only did TCR, not ECS. ECS was constrained based on recent warming by Bengtsson and Schwartz (2013), Jimenez-de-la-Cuesta and Mauritsen (2019) and Nijsse et al. (2020).*

We thank the reviewer for these recommendations, we cite Nijsse et al. 2020 in the revised manuscript (Lines 58, 72).

However, the Tokarska, Hegerl, Schurer, Forster and Marvel "Observational constraints on the effective climate sensitivity from the historical period" (2020) study in ERL does indeed constrain ECS, and so we have kept the reference to Tokarska et al. (2020) as a citation for ECS. Perhaps the reviewer was thinking of the Tokarska, Stolpe, et al (2020) study in Science Advances that does only consider TCR. However, it is the Tokarska, Hegerl et al.(2020) study that constrains ECS that is cited in our manuscript (Lines 578-579)

*53, perhaps delete 'at any given time or timescale'*

Agreed, this is deleted in a revised manuscript.

*58, perhaps worthwhile mentioning those studies that are relying on these models, and why the authors of this study believe their method makes avoiding GCMs for estimating time-dependence is possible? See also major points.*

Agreed. We provide an example of a study that requires complex model output to explore ECS is Nijsse et al (2020), although any study that uses an emergent constraint on ECS or TCR could be used as an example (Lines 57-59):

"Our estimates of ECS and TCR are independent of simulated warming responses in complex climate models (in contrast to estimates utilising complex model output via emergent constraints, e.g. Nijsse et al., 2020)."

The way our method is able to constrain ECS and TCR utilising time-varying climate feedbacks is now explored in the main text (Lines 61-72) and Supplementary Information (Supplementary Section S3, equations S4-S10; Supplementary Figure S7).

*71, 'Quisque' is not a word in my vocabulary. According to wikipedia it is a pre-historic herring."*

Thank you, this word was a typo and does not appear in this revised version.

*81, perhaps nit-picking, but surface albedo feedback, at least that associated with seaice, is not as fast as water vapor, see for instance Tietsche et al. (2011) that find a 1-2year timescale.*

Agreed, it is true that the surface sea-ice albedo component of the fast feedback does strictly have a timescale longer than the residence timescale of water vapour in the atmosphere. We now state that in our revised manuscript (Lines 88-90):

"Also, the snow and ice albedo feedback has a timescale longer than the atmospheric water vapour residence timescale, but is included in $\lambda_{fast}$ here as the timescale snow and sea-ice responds significantly faster than multi-decadal timescales."

*89-90, It would be useful to display the used forcing in a figure, for example to show priors and posteriors of for example aerosol forcing, equivalent to Figure 2.*

Thank you for highlighting this. In our revised manuscript we include a figure showing the prior and posterior distributions of recent radiative forcing (Figure 3), with comparisons to the IPCC estimate and a range of CMIP6 models analysed by Smith et al (2020).

*161, However, very strongly cooling aerosols would result in mid-century cooling because of the different evolutions of aerosol and greenhouse gas forcing (e.g. Stevens2015, Bellouin et al. 2019). Supposedly the bayesian method applied automatically filters out these values, which is why I would like to see the posterior distribution of aerosol forcing.*

We agree that very strongly cooling aerosols would likely result in mid-century cooling. We also agree that our Bayesian approach filters out combinations of greenhouse and aerosol radiative forcing sensitivities that give rise to historic warming trends that are not consistent with observations. In this revised manuscript we have included the prior and posterior distributions of recent aerosol radiative forcing in a new figure (Figure 3).

*"204, Here, I suggest to again remind the reader of the sign convention"*

Agreed, although this particular section of the text is removed we do remind the reader of the sign convention (e.g. Line 92 and Line 249_:

"…noting that the sign convention used implies amplifying feedback from negative $\lambda$."

*218, why not use a doubling of CO2? This is how ECS is defined.*

We agree that a CO2 doubling would also work here. We have chosen to use a 4xCO2 perturbation, in line with one of the standard idealised scenarios for CMIP-class models. Note that WASP does not (yet) have a state-dependence on lambda and so the results of a 2xCO2 experiment would be equivalent (but with a slightly lower signal to noise ratio, where the noise is driven by the imposed interannual variability in Earth energy balance in WASP).

*224, by 90 do the authors mean 5-95?*

Agreed that this was unclear, we now specify the percentile intervals as well as the ranges (e.g. Line 263:

"… varying from 2.1 °C (1.6 to 2.5 °C at 90% range from 5[th] to 95[th] percentiles)"

*248-250, or perhaps a better constraint on total lambda, say based on paleoclimates?*

Agreed – but this section has been re-written due to the differing results now that we have updated to HadCRUT5 temperature reconstructions.

*270, this section added no new information that had not already been provided. I suggest removing it.*
*277, yet these components only explain 1/3 of the total variance?*

We agree our manuscript that lacked clarity on the benefits and motivation behind the principle component and stepwise regression sections (4.2.2 and 4.2.3 respectively). We now explain why the PC analysis and stepwise regression is conducted (Lines 274-279):

"The observational records provide constraints on the parameters of the posterior ensembles that manifest not only as posterior distributions for these parameters but also as relationships between them, as well as between model parameters and key model outputs of interest (such as ECS(t)). While the correlation structure of the 25 parameters' joint posterior distribution is generally quite complex, some key structures emerge that indicate how ECS and TCR uncertainties might be reduced. This method of analysing variation, and simplifying the degrees of freedom of variation, in large data-constrained efficient model ensembles may ultimately help explore parameter space in more complex Earth system models."

With the updated results (moving to HadCRUT5 as the temperature constraint), the first 5 Principle Components now explain 60% of the variance in the posterior model dataset.

*353, this statement requires there are no slow feedbacks acting on timescales from decades to millennia. I recommend to remove this statement, or strongly caveat.*

Agreed, we now state how slow feedbacks acting on many timescales from multi-decadal to millennia may affect how our estimate comnpares to those from palaeoclimate studies (Lines 376-379):

"…but note that additional slow feedbacks not considered here, acting from many decades to millennia, may affect how our estimates are comparable to  estimates of climate sensitivity from the palaeo-record where any longer, (e.g. Rohling et al., 2012; 2018)."

*"368, not 'multiple' but 'two distinct' timescales."*

Agreed we now state that we consider the instantaneous Planck feedback and two further timescales of feedback (now Lines 62-65):

"This study considers the instantaneous Planck feedback and two further timescales of climate feedback: a multi-diurnal feedback representing a selection of fast climate processes, such as water vapour and clouds, and a multi-decadal climate feedback representing slower processes, such as the surface warming pattern effect."

*"370, IPCC 'likely' means 66 percent probability or better."*

Agreed, this is clarified (Lines 394-395):

"…IPCC ECS likely (66% chance or better) …"

*371, for Sherwood et al. (2020) probably the number referred to is 17-83 percent.*

Agreed, this is clarified in this revised manuscript (Lines 392-393):

"…the recent Sherwood et al. (2020) Bayesian review has a narrower baseline 17[th] -83[rd] percentile (66%) range of 2.6 to 3.6 K."

---

## Author Response (AR2)

*The paper has improved in this iteration. A few additional comments before I can recommend publication: I'm particularly interested in a better explanation for the choice of priors, and its influence on the conclusion.*

We thank the reviewer for their careful reading of the manuscript and helpful insight. Below we detail how we have amended the manuscript to address the reviewer's comments.

Major comments
*\* there are a couple of instances where the authors use variable names that deviate from the standard within the literature. For instance, they call their climate sensitivity 'effective climate sensitivity', a term usually reserved for estimates of climate sensitivity for which lambda is assumed constant (so underselling their work). Furthermore, they abbreviate effective climate sensitivity as ECS, which normally stands for 'equilibrium climate sensitivity'. I urge the authors to replace their use of the term ECS with S, with appropriate subscripts (So S_{20 years}, with S_{140} corresponding to ECS). See https://www.ipcc.ch/sr15/chapter/glossary/, under climate sensitivity.*

We agree that our notation was not in keeping with the literature. We have now amended our notation as suggested, such that 'ECS_{140}' has now become 'S_{140}' and so on for all the timescales.

*\* the prior on lambda multidecadal seems to have a strong influence on the conclusions. The main text refers to the supplementary information for explanation about the prior given, but I don't really see much information there either. I disagree with the other reviewer (sorry! I know how annoying disagreeing reviewers are..) that a prior symmetrical around zero is physically justified. Evidence from climate models consistently show that changes in lambda on that timescale have an amplifying effect. The authors indicate they want an estimate of S independent of model evidence, but section 4.2 of Sherwood also summarizes observational and process-based argumentation for a decreasing lambda.*
*\*\* I think the previous asymmetry in the prior was fine. If the authors, torn between reviewers, don't want to revert, I'm happy to see the two priors side-by-side.*

We thank the reviewer for their insight. We now present the original prior for multi-decadal climate feedback (symmetrical about zero) alongside an alternative prior that is not symmetrical about zero and derives from Sherwood et al (2020)'s analysis of the pattern effect in section 4.2 therein.

This is discussed in a new sub-section 4.3 (Lines 405 to 435) in a new Figure 8, and in a new column of Table 1.

We also now discuss the implications of these results with an alternative prior in the conclusion (lines 481 to 485 and lines 501 to 505).

*\*\* Please provide a paragraph of explanation of prior choices in the main text*
*\* the authors use a uniform prior for lambda multidecadal and lambda fast, which is typically not recommended (see the STAN user manual: https://github.com/stan-dev/stan/wiki/Prior-Choice-Recommendations). For consistency, the authors should use the same distribution for lambda Planck, lambda fast and lambda multidecadal (normal distribution seems fine, as the overall feedback is the sum of local feedbacks). There is no physical reason to say that lambda is necessarily between the two given boundaries, and the fact that the posterior does not drop to zero at the boundary indicates the data may indeed by consistent with lambdas < 3.*

We now discuss the choice of priors in the new sub-section 4.3 (Lines 405 to 435). Essentially, we have relatively high confidence in the Planck feedback, and so adopt a normal prior with relatively low uncertainty. However, we adopt a position of ignorance about lambda_fast and lambda_multidecadal and so use a uniform distribution for these in the standard case. The reason lambda_fast and lambda_multidecadal have a lower end of -3 Wm-2K-1 is that we know that the value of each must be greater than minus lambda_{Planck}, explained on lines 411 to 413. This is because the total lambda must always be positive on any timescale.

If we had not imposed this then we could see a situation where lambda_{Planck} + lambda_{fast} was less than zero (and so non-physical) but the model would still work mathematically, provided lambda_{multidecadal} were large enough, due to the finite timestep of the model – event though this would be a non-physical situation in reality because on smaller timesteps than the model resolves total lambda would be negative (and so non-physical in our sign convention). Therefore the hard limit of -3Wm-2K-1 is imposed for both lambda_fast and lambda_multidecadal (where lambda_Planck = +3.3±0.2 Wm-2K-1).

Both lambda_fast and lambda_multidecadal are extended far enough in the positive direction to be past the point where the observation-consistent simulations have already dropped to zero.

Minor comments:
*multi-diurnal -> multiday?*

Agreed, we have now changed multi-diurnal to multiday (Line 71).

*line 380: incomplete sentence*

Agreed, we have now completed the sentence which reads (Line 460):
        "Note that additional slow feedbacks not considered here, acting from many decades to millennia, may affect how our estimates are comparable to estimates of climate sensitivity from the palaeo-record where any longer feedbacks have been treated as radiative forcing (e.g. Rohling et al., 2012; 2018)."